# Policy Learning with a Language Bottleneck

**Megha Srivastava**  *megha@cs.stanford.edu*
*Stanford University*

**Cédric Colas**  *ccolas@mit.edu*
*Massachusetts Institute of Technology*
*Inria*

**Dorsa Sadigh**  *dorsa@cs.stanford.edu*
*Stanford University*

**Jacob Andreas**  *jda@mit.edu*
*Massachusetts Institute of Technology*

**Reviewed on OpenReview:** *https://openreview.net/forum?id=sK8uEqzQPv*

## Abstract

Modern AI systems such as self-driving cars and game-playing agents achieve superhuman performance. But they often lack human-like generalization, interpretability, and inter-operability with human users. This paper introduces *Policy Learning with a Language Bottleneck* (PLLB), a framework enabling AI agents to generate linguistic rules that capture the high-level strategies underlying rewarding behaviors. PLLB alternates between a *rule generation* step guided by language models, and an *update* step where agents learn new policies guided by rules. Crucially, PLLB enables this kind of language-guided learning even when a natural language rule is insufficient to completely describe the target policy. Across five diverse tasks, including a two-player signaling game, maze navigation, image reconstruction, and robot grasp planning, we show that PLLB learns more interpretable and generalizable behaviors than standard policy learning methods. In three additional human subject studies, we show that show the learned rules significantly improve *human* task performance, enabling more effective human-AI coordination.[1]

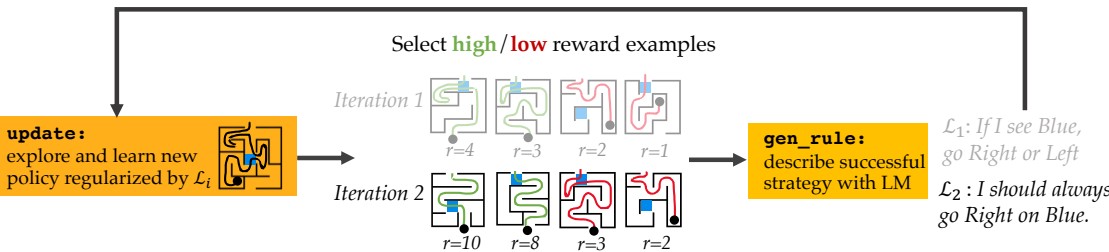

Figure 1: *Policy Learning with a Language Bottleneck* (PLLB) alternates between two steps: 1) `gen_rule` generates a linguistic rule $\mathcal{L}_i$ explaining the agent's best behaviors by prompting a language model with contrastive (positive and negative) episodes; 2) `update` learns a new policy conditioned on $\mathcal{L}_i$. PLLB strengthens human-AI coordination by constraining policies to be more interpretable and generalize better.

---

[1] We provide source code for our experiments at https://github.com/meghabyte/bottleneck.

# 1 Introduction

As AI systems play an increasingly central role in automation and decision-making, their success depends not only on their performance, but also on their ability to model and align with human behavior. To be truly effective assistants, these systems must act and generalize in ways that align with human expectations. However, many of today's AI systems do not meet these standards. Self-driving cars or game-playing agents like AlphaZero may achieve super-human performance, but they lack interpretability (McIlroy-Young et al., 2020) and often act unpredictably, especially outside their training distribution (Wang et al., 2023).

Unlike AI systems that are trained in isolation, humans acquire most of their skills and knowledge from interacting with others, often using language — via instructions, advice, or explanations that improve their decision-making capabilities (Carruthers & Boucher, 1998; Mesoudi & Thornton, 2018). Language acts as a *communicative* medium, enabling us to teach, learn from, and coordinate with others to solve complex problems. It also supports other *cognitive* functions: even when not used for communication, it allows us to represent abstract concepts (Hesse, 1988; Lakoff & Johnson, 2008), and plan (Vygotsky, 1965; Clark, 1998); it guides our attention (Waxman, 1994; Yoshida & Smith, 2003), and prompts relational thinking (Gentner & Loewenstein, 2002).

Consider a driver learning to navigate novel social conventions (e.g., triangle-shaped stop signs). While adapting to the environment, they might verbalize strategies to themselves to avoid future mistakes (e.g., *If the sign is triangular, I should stop*, a cognitive use), or transmit this convention to others (e.g., telling a friend *In Japan, stop signs are triangles*, a communicative use). Representing learned information in language helps humans solve problems and transmit knowledge by effectively capturing abstract problem structures that facilitate learning and generalization (Boutonnet & Lupyan, 2015; Chopra et al., 2019; Tessler et al., 2021). Importantly, language remains useful even when it cannot fully encapsulate an entire strategy (e.g., a drivers' reflexive actions, or fine-grained driving mechanics).

There exist many recent examples in the literature of AI systems leveraging language-based representations or linguistic feedback, but these often rely on external supervision: humans in the loop or hard-coded feedback functions (Luketina et al., 2019; Colas et al., 2022). We argue that more human-like AI systems should not only *use* language-based supervision but also *generate their own language-based feedback* to leverage both the communicative and cognitive functions of language. Importantly, this capability should also extend to tasks that require updating an underlying policy that is only partially expressible in natural language, such as low-level control or reflexive actions.

This paper introduces *Policy Learning with a Language Bottleneck* (PLLB), a framework that provides artificial embodied agents the ability to generate linguistic rules that capture the strategies underlying their most rewarding behaviors. As shown in Figure 1, PLLB alternates between a `rule generation` step that explains the agent's experiences by prompting a language model (LM) with contrastive episodes, and a `policy update` step that learns a new policy guided by these rules. Unlike past work that solely leverages LMs for modeling agent behavior and multistep reasoning (Park et al., 2023; Wei et al., 2023), PLLB is applicable even when aspects of the target policy cannot be expressed with language.

*Policy Learning with a Language Bottleneck* can be applied to a wide range of agent types, from RL policies to LLM-based learners to robot pose estimators, and the core mechanism of PLLB remains the same: using contrastive reward signals to extract linguistic rules via an LLM, and then conditioning policy updates on these rules depending on the type of learning agent. We investigate the role of PLLB in shaping more human-like policies across five distinct tasks. **They perform better**: in two image reconstruction tasks, PLLB agents generate instructions increasing the listeners' performance compared to non-linguistic baselines (Section 7), and PLLB agents also help more efficiently learn robot grasping policies (Section 8). **They improve few-shot generalization**: in a maze task, PLLB rules uncover abstract problem structure that improve learning similar mazes (Section 6), and PLLB also reduces reliance on non-generalizable visual features in robotic manipulation (Section 8). **They are more interpretable**: in a coordination task, agents converge on humans' preferred policy when multiple optimal policies exist (Section 5). **They are more inter-operable**: in maze and image reconstruction tasks, humans achieve better rewards when interacting with PLLB agents compared to agents trained without a bottleneck (Sections 6 and 7).

## 2 Background & Related Work

PLLB is inspired by the dual use of language as both a communicative and cognitive tool for decision-making in both humans (Carruthers & Boucher, 1998) and machines (Colas et al., 2022).

**Language for communication.** Language facilitates cooperation and coordination between humans and machines via instructions (Hermann et al., 2017; Chevalier-Boisvert et al., 2019), advice (Watkins et al., 2021), explanations (Zhong et al., 2020; Lampinen et al., 2022), or the formation of conventions (Hawkins et al., 2020; Hu & Sadigh, 2023). Such communicative functions increase the fidelity and breadth of cultural transmission — a process of social learning that underlies human ecological success (Mesoudi & Thornton, 2018). PLLB agents not only learn from language, but also generate their own to be shared with others.

**Language for cognition.** Language also augments a learner's cognitive abilities. Language-augmented RL agents represent more abstract goals (Jiang et al., 2019), generalize better (Hill et al., 2020; Colas et al., 2020; Wong et al., 2021), explore more efficiently (Colas et al., 2020; Tam et al., 2022; Klissarov et al., 2023) and can decompose complex goals into simpler ones (Chen et al., 2021; Ahn et al., 2022; Hu et al., 2022; Sharma et al., 2021; Hu & Clune, 2023). Our work extends these benefits to agents that learn from *self-generated* linguistic feedback.

**Inner speech.** Generating linguistic rules for oneself is a form of *inner speech*. In the Vygotskian tradition, inner speech is seen as the internalization of the social speech generated by caretakers to help children solve problems (Vygotsky, 1965; Luria, 1959). As a result, it is thought to support our capacities for complex, long-term behaviors (Vygotsky, 1965; Luria, 1959; Hermer-Vazquez et al., 2001; Spelke, 2003). AI agents endowed with forms of inner speech (explanations, descriptions or subgoals) have been found to perform and generalize better than agents trained with purely neural representations (Wong et al., 2021; Lampinen et al., 2022; Roy et al., 2022; Hu & Clune, 2023; Kim et al., 2020). Unlike these approaches, PLLB approach generates language in an unsupervised way and maximizes downstream performance as well as interpretability and inter-operability.

**Multi-step reasoning and text agents.** Recent works have proposed guiding LMs' reasoning by prompting them to step through sequences of "thoughts" (Wei et al., 2023; Yao et al., 2022; Li et al., 2023b; Shinn et al., 2024). Our approach similarly uses a language bottleneck to concisely express intermediate information useful for later behavior (i.e. policy learning). However, unlike text-based reasoning agents such as ReAct (Yao et al., 2022), Rememberer (Zhang et al., 2023), and Reflexion (Shinn et al., 2024), **PLLB does not require the underlying policy to be fully expressible in text**. For example, we show that a text-only maze solving agent completely fails, whereas PLLB can improve classical Q-learning style agents (Section 6). Moreover, instead of describing actions for a particular state, PLLB rules capture high-level strategies over a sequence of states and actions, enabling generalization. We also show the learned rules can be shared to improve human task performance across multiple user studies (Sections 6 and 7).

## 3 The Language Bottleneck

PLLB builds on the standard RL framework to train agents to solve decision-making tasks in human-like ways. We formalize decision-making tasks (e.g., solving a maze) as Markov decision processes $(S, A, f, R, T)$ with reward function $R : S \times A \to \mathbb{R}$ (e.g., solving speed) over states $S$ (e.g., cell coordinates) and actions $A$ (e.g., directions), finite time horizon $T$, and a deterministic transition function $f : S \times A \to S$ that maps state–action pairs $(s, a)$ to next states $s$ (e.g., moving between cells). Standard RL then involves training a policy $\pi$ to maximize expected reward (e.g., learning to solve the maze faster). This is usually done by alternating between two steps: (1) collecting data with the current policy: $D \leftarrow \pi_i$ and (2) updating the policy using the data: $\pi_{i+1} \leftarrow \texttt{update}(\pi_i, D)$, where $\texttt{update}$ implements an RL algorithm (Sutton & Barto, 2018). This procedure often fails to yield interpretable, human inter-operable, or generalizable behaviors.

Our key idea is to introduce a *language bottleneck* between data collection and policy update. We extract linguistic rules explaining past rewarding behaviors and then use them to regularize the policy's behavior in the next learning iteration (e.g., *I should go right in blue cells*, see Figure 1). Our experiments will show that this improves the interpretability and inter-operability (communicative use), and performance and

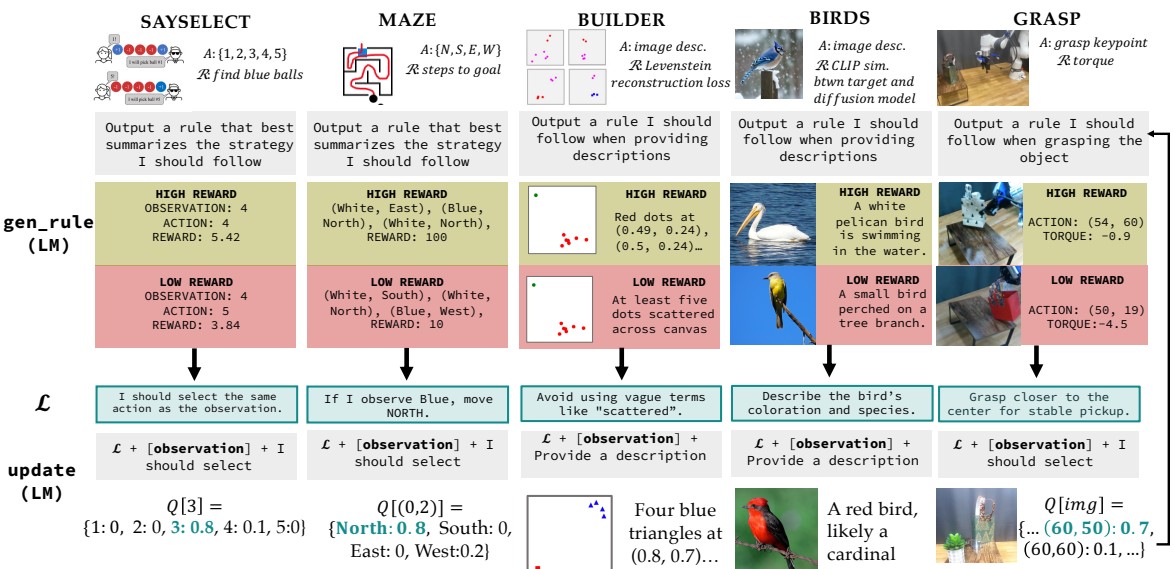

Figure 2: PLLB can be applied to a diversity of domains. In all domains, it iterates between `gen_rule`, a function that prompts an LM (top gray boxes) to extract a linguistic rule ($\mathcal{L}$, blue) by contrasting high and low reward episodes from past experience (green and red boxes), and `update`, a function for updating an agent's policy through interaction with the environment conditioned on $\mathcal{L}$. PLLB can be applied to multi-step decision making tasks and visual or robotics tasks alike with minimal implementation variations (replacing the LM by a VLM in `gen_rule`, or replacing policy regularization by simple instruction conditioning in `update`. Gray boxes represent prompts `gen_rule` and `update`, and full prompt details are in Appendix Section B.7.

generalization (cognitive use) of our agents over standard RL baselines in a variety of tasks (see Figure 2 and Sections 5 to 7). The resulting algorithm thus alternates between three steps: (1) data collection, $D_i \leftarrow \pi_i$, which is implemented just as in ordinary policy learning approaches; (2) language bottleneck generation, $\mathcal{L}_i \leftarrow \texttt{gen\_rule}(D_i)$ and (3) policy updating, $\pi_{i+1} \leftarrow \texttt{update}(\pi_i, D_i, \mathcal{L}_i)$ (see Figure 1).

The core mechanism underlying PLLB is consistent across all instantiations: `gen_rule` prompts a language model with *contrastive* examples (high- vs. low-reward trajectories) to extract a rule $\mathcal{L}$ capturing what distinguishes successful from unsuccessful behavior, and `update` conditions the agent's subsequent learning on $\mathcal{L}$. What varies across domains is only how `update` incorporates the rule, which is depends on the learning agent's modality in that domain (e.g. Q-value regularization (Sections 5–6), prompt conditioning for LMs (Section 7) or visuomotor control (Section 8). We next describe these the `gen_rule` and `update` steps in more detail.

## 3.1 Rule Generation (`gen_rule`)

Using all the experience $D_i$ collected by the policy $\pi_i$ in the current iteration, `gen_rule` aims to infer an abstract rule $\mathcal{L}_i$ that best explains the agents' successful behaviors: $\mathcal{L}_i \leftarrow \texttt{gen\_rule}(D)$. This is done by prompting an LM with contrastive episodes from $D_i$ (top-$N$ highest vs. top-$N$ lowest total rewards) and asking it to *provide the rule that should be followed to obtain high rewards* (see first row of Figure 2 and Appendix Section B.7 for full prompts). Importantly, this requires the first iteration of `gen_rule` to start only once we observe a pair episodes with sufficiently different rewards. We found this contrastive approach, inspired by Zhong et al. (2023) and Dunlap et al. (2023), to provide more precise rules than simply summarizing high-reward strategies.

### 3.2 Rule-Guided Policy Update (`update`)

Given a rule $\mathcal{L}_i$, the `update` step produces a new policy $\pi_{i+1} \leftarrow \texttt{update}(\pi_i, D_i, \mathcal{L}_i)$ that is better aligned with $\mathcal{L}_i$. There exist many methods for leveraging language instructions to update agents' policies, though these methods traditionally focus on instructions provided by human experts. Crucially, which method to implement PLLB with depends on the underlying agent and action space representation, and will improve with advances in multi-modal modeling.

For RL policies, we leverage InstructRL (Hu & Sadigh, 2023), which regularizes the learned policy with another policy induced by the linguistic rule $\pi_{\mathcal{L}}$. For instance, in the maze example shown in Figure 1, if the rule is *I should go right on every blue cell*, the induced policy $\pi_{\mathcal{L}}$ should assign probability 1 to the *right* action in blue cells, but equal probabilities to all actions in every other situations. In the Q-learning algorithm, this approach simply adds a *regularizing term* (orange) to the standard Q-learning update rule [2]:

$$Q^\theta(s_t, a_t) \leftarrow r_{t+1} + \gamma Q^\theta(s_{t+1}, a_{t+1}) \quad \text{where} \quad a_{t+1} = \arg\max_a \left[ Q(s_{t+1}, a) + \lambda \log \pi_{\mathcal{L}}(a \mid s_t) \right].$$

Here $\gamma$ is a discount factor and $\lambda$ controls the strength of the rule-induced regularization and could be made time-dependent with a pre-defined or learned schedule (e.g., stronger regularization early).

But how do we induce $\pi_{\mathcal{L}}$ from $\mathcal{L}$? Since $\mathcal{L}$ is expressed in natural language, we may obtain a rule-conditional policy by prompting an LM. In particular, our experiments condition the LM on both the current rule and the current state of the agent $s_t$, and instruct the LM to generate the next action to obtain a probability distribution over admissible actions (e.g directions), as in Hu & Sadigh (2023). While these prompted LMs may perform tasks poorly on their own (e.g. because of their inability to perform long-range planning or process complex visual input), the regularizer may nonetheless guide Q-learning in the right direction. Running Q-learning updates with the regularization term from experience data gives us a new Q-table $Q_{i+1}^\theta$ from which we can derive the new policy $\pi_{i+1}$ by taking actions with maximum expected value in every state $\pi_{i+1}(s_t) = \max_a Q_{t+1}^\theta(s_t, a)$.

Finally, there exist some domains where the policy can be directly implemented by an LM or VLM, such as text-based games explored by methods like ReAct (Yao et al., 2022) or Rememberer (Zhang et al., 2023). In these cases, `update` can be implemented by conditioning the policy on the rule $\mathcal{L}$ and adding it to the prompt $\pi_{\mathcal{L}}$, ultimately steering the agent's: $\pi_{i+1} \leftarrow \pi_{\mathcal{L}_i}$. Given a new policy $\pi_{i+1}$, we can now close the loop and generate a new rule $\mathcal{L}_{i+1}$. The choice between InstructRL-style regularization and prompt conditioning depends on the action space and policy architecture. In settings with large, continuous action spaces (e.g. reasoning over free-text), conditioning text-generation on rules is likely more tractable, whereas Q-learning style updates may be more appropriate for interactive environments with feedback (e.g. game playing). We implement PLLB with tabular RL agents in Sections 5 and 6, with LM/VLM policies in Section 7, and visuomotor robot controllers in Section 8.

## 4 Experiment Set-Up

We run experiments on a diverse set of five tasks, with different agent architectures and action spaces, to showcase how PLLB can train more human-like agents. These include a simple two-player communication game called SAYSELECT (Section 5), a MAZE solving task (Section 6), two collaborative image reconstruction tasks with synthetic (BUILDER) and natural (BIRDS) images (Section 7), and grasp planning for a 7-DoF robot (Section 8). For all tasks, we include hyperparameter details (Appendix B.1), the exact prompts used in `gen_rule` and `update` (Appendix B.7), and examples of generated rules. For our main experiments, we prioritized using accessible and low-cost open weight language models such as `llama-2-70b-chat`. However, we analyze the effect of changing the underlying language model on `gen_rule` in Appendix B.8 and B.9.

To evaluate the *cognitive* function of PLLB, we first evaluate the policies learned by artificial agents in each environment based on performance metrics such as task reward, similarity to known human-like policies, or

---

[2]This regularizing term can also be applied in the function approximation setting (Mnih et al., 2013).

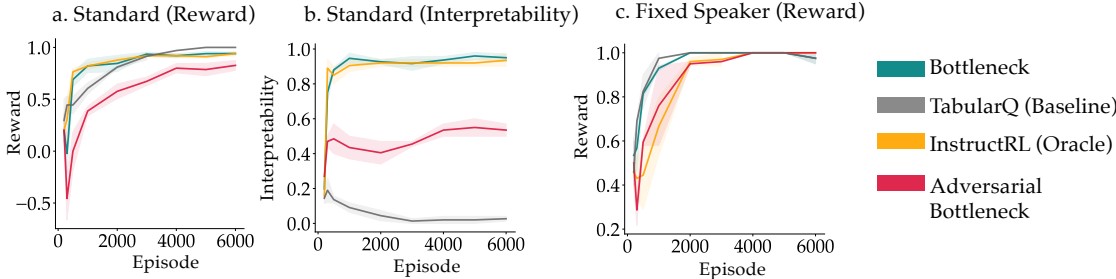

Figure 3: SAYSELECT. a) **Bottleneck** agents learn as fast as **TabularQ** and **InstructRL** and faster than agents using adversarial rules **Adversarial**. b) Unlike **TabularQ**, they learn human-interpretable policies; this without relying on an external instruction like **InstructRL**. c) When faced with speakers enforcing a non-human-interpretable policy, **Bottleneck** converges faster than **InstructRL**.

generalization. For all tasks, we compare our **Bottleneck** approach with baseline methods (e.g., standard RL, base LM) and an **Adversarial** ablation of PLLB that leverages invalid linguistic rules (e.g., obtained by labeling high reward examples as low reward) in `gen_rule`. The purpose of **Adversarial** is to test whether PLLB gains arise from meaningful rule content rather than merely introducing any linguistic signal.

To evaluate the *communicative* function of PLLB, we conduct several human subject studies showing that generated rules $\mathcal{L}$ can help improve human task performance on a variety of metrics (e.g., reward, solving time). Participants were either recruited in-person (for MAZE) or on Prolific (BUILDER, BIRDS), compensated at a rate of US$12/hour. All studies were approved by our institution's IRB. To the best of our knowledge, our work is the first to validate self-generated agent rules from LMs for training human participants.

Finally, we conduct additional sensitivity analysis on the affects of variations in prompt, model temperature, and non-contrastive episode selection in the Appendix.

## 5 SaySelect

As a proof-of-concept, we first consider SAYSELECT, a simple collaborative game introduced in Hu & Sadigh (2023). Here, a *speaker* can see a hidden set of five balls including two blue ones and must help a *listener* find the two blue balls by communicating with them only via numbers, see Appendix Figure 8. In principle, the speaker and listener could converge on any bijective convention associating messages to balls, and this is what we find with RL agents (multiple optimal policies). Humans, on the other hand, empirically prefer the mapping $1 \rightarrow 1$, $2 \rightarrow 2$, etc. which we call the *human-interpretable policy*. Whereas Hu & Sadigh (2023) showed that regularizing the listener's policy with the instruction *I should select the same number as my partner* successfully guided the two RL agents to the human-interpretable policy, we ask whether PLLB could learn a similar rule and achieve a similar interpretability *on its own*?

**Task Overview.** The speaker and listener are RL agents trained with Q-Learning that receive positive rewards when the listener collects blue balls, negative rewards otherwise. After training both agents for 200 episodes, we prompt llama-2-70b-chat[3] with the task description and a contrastive set of high- and low-reward episodes to generate a rule $\mathcal{L}$, e.g., *I should choose action 4 whenever the observation is 2, 3, 4, or 5.* (`gen_rule`, see Figure 2).

We generate new rules every 500 training episodes. In between, we regularize the listener's policy with a another policy $\pi_{\mathcal{L}}$ induced from the current rule $\mathcal{L}$. This is done by applying a *softmax* on the LM's logits (see method in Section 3) across all possible actions and obtaining a distribution. In the following experiments, we compare our **Bottleneck** method, with the **Adversarial** version (corrupted rule), a **TabularQ** baseline, as well as an **InstructRL** upper bound using the same predefined instruction as (Hu & Sadigh, 2023).

---

[3]https://huggingface.co/meta-llama/Llama-2-70b-chat-hf

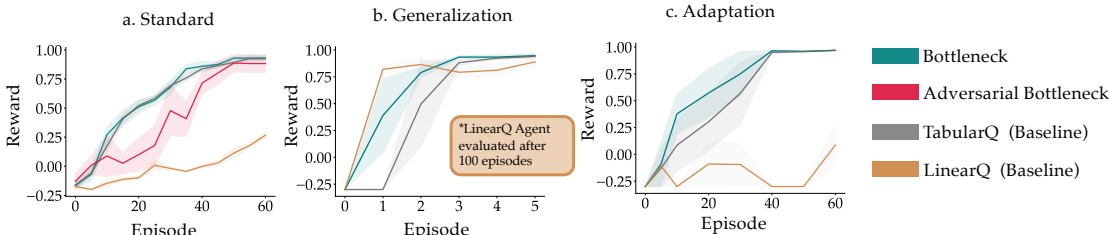

Figure 4: Results in MAZE. a) **Bottleneck** agents learn as fast as the non-linguistic **Baseline agents**, but faster than **Adversarial** and **LinearQ** agents. b) When faced with a new maze with similar structure, **Bottleneck** agents learn faster than **TabularQ** (which does not perceive color) and **LinearQ** (which does, but learns slower). c) When faced with a maze of a different structure, **Bottleneck** agents adapt swiftly while **LinearQ** cannot recover. We do not evaluate **Adversarial** for the Generalization or Adaptation experiments due to its poor performance in the standard setting.

## 5.1 PLLB helps learn human-interpretable policies

We first compare the four methods along two dimensions: their reward and their *interpretability* measured as the proportional (0 to 1) similarity between the listener's policy and the optimal human-interpretable policy (the intuitive message-ball described above). All methods but **Adversarial** quickly solve the task (Figure 3a), showing that corrupted rules hinder learning. **Bottleneck** and **InstructRL** both converge on the human-interpretable policy thanks to their linguistic rules while **TabularQ** — deprived of such rule — does not (Figure 3b). Interestingly, **Adversarial** policies are more interpretable than **TabularQ** ones, suggesting that even corrupted language may provide inductive biases towards more human-like behavior. Qualitatively, we observe that generated rules converge towards describing the human-interpretable policy (e.g., *I should follow the strategy of choosing the same action as Agent 1*, see examples in Section B.8).

## 5.2 PLLB can learn counter-intuitive policies

A potential confound in the preceding experiment is the possibility that PLLB converges to a human-interpretable policy because this is the *only* behavior an LM can describe. To test this hypothesis, we fixed the speaker to use a counter-intuitive policy using a random mapping between balls and messages (e.g., $1 \rightarrow 3$, $5 \rightarrow 2$, etc.). Here, the human-interpretable policy is mis-aligned, and would lead the listener to fail or learn slower. In Figure 3, we observe that while **InstructRL** and **Adversarial**, both regularized by misaligned rules (the human-interpretable one for **InstructRL**, another random permutation for **Adversarial**), eventually adapt to the fixed speaker, they converge at a slower rate than both **TabularQ** (no rule) and **Bottleneck**. This effect was more pronounced at higher values of $\lambda$. Overall, **Bottleneck** is the only method that adapts to a fixed speaker policy while also converging on the human-like policy when the speaker is learning, without human supervision.

## 6 Maze

In the MAZE domain, agents must navigate a maze to find the goal using four directional actions (N/S/E/W). We study mazes in which environment cues provide hints about the optimal path. This setup helps us study the impact of the language bottleneck: can PLLB agents infer the underlying structure and use it to generalize across mazes (cognitive use)? Can they transmit this knowledge to humans and help them perform better in novel settings (communicative use)?

**Task Overview.** We randomly generate different 7x7 mazes with the gym-maze code base.[4] Agents receive as reward the inverse of the number of steps they needed to reach the goal as an incentive to learn the optimal solution. Although all mazes are random, we introduction additional structures by coloring cells for which the optimal action is *south* (red) or *north then east* (blue) with probability 50% (or they are left blank).

---

[4]https://github.com/MattChanTK/gym-maze

Therefore, agents that leverage color information can better generalize to new mazes where the coloring is preserved, even if the optimal action sequence differs.

The first iteration of PLLB corresponds to training a standard RL algorithm to obtain $\pi_1$, in our case a tabular Q-learning agent (**TabularQ**). After the agent observes at two solved mazes, we run `gen_rule` by prompting a llama-2-70b-chat LM with contrastive episodes and a task description. This gives us a linguistic rule $\mathcal{L}_1$ (e.g., *Upon observing RED, take SOUTH*) that we can use to `update` the policy. We first induce the regularizing policy $\pi_{\mathcal{L}_1}$ with rule $\mathcal{L}_1$ by obtaining a probability distribution over the 4 actions from the LM, and then run the RL algorithm for 5 episodes to obtain the new policy $\pi_2$ (procedure described in Section 3). We repeat these steps every 5 episodes of interactions with the environment.

Baselines include the tabular Q-learning algorithm without language bottleneck (**TabularQ**) and a variant of **Bottleneck** generating rules from reward-randomized episode samples (**Adversarial**). We also evaluate **LinearQ**, an agent learning a linear model Q-function with an additional feature for cell color. We train **LinearQ** with a batch size of 10 and learning rate 0.001, after performing a hyperparameter sweep. We found **LinearQ** took significantly longer to converge than the other methods, likely due to the increased state representational complexity.

## 6.1 PLLB improves few-shot generalization

We next evaluate PLLB's ability to improve the few-shot generalization capabilities for policies on unseen mazes with a similar underlying structure. For a a fair comparison, we start with the fully converged policy for each method (requiring 100 episodes for **LinearQ**). Figure 4a shows that, in the Standard setting, learning a valid rule (**Bottleneck**) does not increase learning speed of a single maze over **TabularQ**. However, using a corrupted rule (**Adversarial**) or learning from linear features (**LinearQ**) slows down learning. Furthermore, **Bottleneck** outperforms **TabularQ** with respect to few-shot generalization: when we switch to a new 7x7 maze sharing the same underlying color semantics, but with a different optimal action sequences. **Bottleneck** leverages the learned rule and generalizes to the new maze more effectively than all other agents (Figure 4b). While **TabularQ** cannot generalize because it does not perceive colors, **LinearQ** does generalize faster at first, but converges slower.

The generated rules improve over time to better capture the underlying structure of the maze (e.g., *if I observe BLUE, then take the NORTH action*; see other examples in Appendix B.8). Across all trials, 100% of the final rules mention the red → *south* rule and 60% uncovered the more complex blue → *north-then-east* rule. Finally, we find that a policy that does not update Q-values (that is, purely implemented via the LM) achieves a success rate 0% within the same time limit, even when using the final rules generated from PLLB. This highlights the importance of leveraging experience that cannot be expressed with language, which text-based reasoning agents cannot do.

## 6.2 PLLB learns adaptable policies

In Figure 4c, we reproduce this same experiment on a maze with a different underlying structure (red now indicates *west* while blue indicates *east then south*). Although the rule **Bottleneck** learned does not apply anymore, it can still adapt faster than the baselines. We find that all trials end with rules capturing the new mapping (100% red → *west*, 50% blue → *east-then-south*). Meanwhile, **LinearQ** overfits to the first maze and struggle to adapt. Overall, our results show that PLLB supports human-like cognitive functions by finding a tradeoff between the efficiency of TabularQ and the generalization capabilities LinearQ, while remaining more adaptable.

## 6.3 PLLB is more interpretable and inter-operable

Can generated rules be useful to humans as well? We asked 50 university students to solve a 7x7 maze in the fewest steps possible using arrow keys. They could only observe the cells they had already visited and the walls they had already bumped into (see Figure 10 in Appendix). We split them into three groups: a **Control** group receiving no assistance, and two others receiving information about a *similar* but different maze sharing the same underlying color semantics. The **Visual** group is shown a visual representation of

the optimal policy (arrows in each cell, see Figure 9 in Appendix), while the **Bottleneck** group is provided a randomly sampled language rule $\mathcal{L}$ learned by PLLB. This set-up lets us evaluate how generalizable the two different aids are to a new maze, as well as how quickly participants can account for incorrect information depending on modality.

Participants using PLLB rules solve the new maze with fewer steps than others and find this aid significantly more useful than the visual one in average (see Figure 5). Participants indeed found it harder to extract the relevant information from the visual aid, which contained a great deal of non-transferable information (optimal actions in all non-colored cells). PLLB rules focus on the useful and transferable information learned by the previous agent, which is much more readily usable by humans. Overall, the results in MAZE demonstrate the ability of PLLB to train agents that are more generalizable and adaptable (cognitive use) as well as more interpretable by, and inter-operable with humans (communicative use).

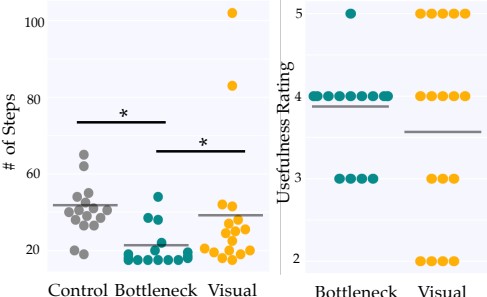

Figure 5: Participants given PLLB rules solve new mazes faster than those given either visual or no aid (control). They also self-reported linguistic feedback as more useful than visual. * marks statistical significance at level 0.05 with two-sided Mann-Whitney test after Bonferroni correction.

## 7 Collaborative Image Reconstruction

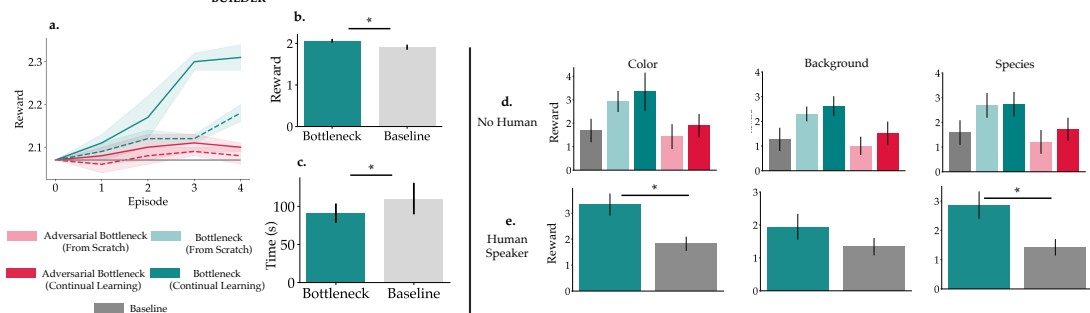

Figure 6: BUILDER: **Bottleneck** listeners improve upon **Baseline** descriptions over time, unlike an (**Adversarial**) listener that inaccurately represents rewards. Human users reconstructed target images with b. higher accuracy, and c. less time, when provided **Bottleneck** instruction vs. original **Baseline** descriptions. BIRDS: PLLB rules (3 iterations) help d. automated and e. human speakers generate better descriptions for naturalistic image reconstruction for three distinct reward functions.

PLLB is not restricted to the training of RL agents. This section introduces two collaborative image reconstruction tasks inspired by the collaborative assembly task of McCarthy (2020). BUILDER and BIRDS both consider two agents: a *speaker*, who can see a hidden target image, and a *listener*, who must accurately reconstruct the target image based on a description from the speaker. Both agents aim to converge to a description style leading to high reconstruction accuracy. We consider both synthetic images built from sequences of discrete actions by the listener (BUILDER), and natural images of birds that the listener reconstructs using a text-to-image generation model (BIRDS). Here, PLLB combines with state-of-the-art multi-modal models to let agents perceive semantic visual features.

### 7.1 Task Overview

**Dataset.** In BUILDER, we construct a dataset of synthetic images with a variable number of shapes (triangle, square, circle) in different colors (magenta, blue, red, green) and locations on a 2D grid. Each image is created by a sequence of discrete actions, each representing a particular combination of shape, color, and

location (e.g., `[ACT12 ACT2 ACT4]`, where ACT2 adds a magenta circle in the lower right area of the grid). In BIRDS, we construct a dataset of natural bird images by selecting 5 images for each of 10 bird species from the CUB-200-2011 dataset provided by Wah et al. (2011) (see Appendix Section B.4 for species list).

**Reward.** In BUILDER, since target and reconstructed images are fully determined by their action sequences, we measure task success via the Levenshtein similarity between the (sorted) corresponding sequences. In contrast, the BIRDS listener outputs text-to-image generations with varied properties, so we define three reward functions: **Color**, **Background**, and **Species**. To evaluate a reconstructed image $x_r$ against a target $x_t$, we form a *contrast* set $C$ from three CUB-200-2011 images differing in the target property (e.g., color) but most similar in others. Reward is then $r = \sum_{c \in C} d(\mathsf{CLIP}(x_r), \mathsf{CLIP}(x_c)) - d(\mathsf{CLIP}(x_r), \mathsf{CLIP}(x_t))$, where $\mathsf{CLIP}$ denotes the CLIP image embedding (Radford et al., 2021). For both BUILDER and BIRDS, rewards are reported on a held-out validation set.

**Models.** As open-source VLMs struggle with describing synthetic images, we use the llama-2-70b-chat LM as our speaker for BUILDER, representing images in raw text that list all shapes' type, color, and exact coordinates individually in sequence. On the other hand, we use the open source Llava VLM as our speaker for BIRDS (Liu et al., 2023). We prompt both speaker models using the target image and a general task description, as well generated rules after the first iteration.

We implement the listener for BUILDER with a neural sequence-to-sequence model pre-trained on English text (Lewis et al., 2019), which we fine-tune at each episode on a training set of (description, action sequence) pairs using descriptions provided by the speaker. Meanwhile, we use the Stable Diffusion text-to-image diffusion model as the listener for BIRDS.

For both tasks, we consider two settings: **From Scratch**, where the listener is initialized using the original pre-trained model weights at every episode, and **Continual Training**, where the listener is continuously updated over time using descriptions by the speaker. We create separate held-out splits for training, early stopping, selecting samples for `gen_rule`, and evaluation.

**Implementation.** We first generate base image descriptions for both tasks by providing the speaker a prompt containing a general task description and target image, but without any rule $\mathcal{L}$. For the BUILDER task, an original image description is: *At least five dots in total - four red and one green - scattered across the canvas, with two sets of matching locations (x = 0.49 / y = 0.24 and x = 0.5 / y = 0.24), another set at x = 0.52 / y = 0.3, and finally, one dot located near x = 0.76 / y = 0.71.* This description is overly complex, mixes different coordinate formats, and difficult for the listener to use to reconstruct the target image, leading to low reward. Meanwhile, an example original image description for the BIRD task is: *"a bird on a branch"*. While simple, this description does not contain sufficient information about the **Color**, **Background**, and **Species**, causing the listener to generate an image reconstruction with low reward.

We next evaluate these base descriptions using each task's respective reward functions, and select the (image, description) examples with the 5 highest and lowest rewards. Using the prompts shown in Section B.7, we implement `gen_rule` using the llama-2-70b-chat and llava models for BUILDER and BIRDS, respectively. For BIRDS, we observe that output rules $\mathcal{L}$ are specific to the reward function: an example rule we get for the **Species** reward is: *"Identify the bird's species if possible, and include any distinctive characteristics that set it apart from other birds."*, while an example rule we get for the **Background** reward is: *"Describe the setting or background of the image, such as the presence of snow, water, or other elements"*.

We implement `update` for both tasks by simply appending the output rule $\mathcal{L}$ to the original prompt to the speaker, as shown in Figure 2, using the speaker's output as the new description for a given target image. We repeat this cycle of eliciting rule $\mathcal{L}$ , appending $\mathcal{L}$ to the same prompt used to generate the base instructions in order to re-label our data with new instructions, and training and evaluating the listener model for a total of 5 iterations for BUILDER and 3 iterations for BIRDS. We qualitatively show this leads to improved image reconstruction from the listener in Appendix Figure 13, and next discuss our experimental results.

## 7.2 PLLB helps speakers provide more usable instructions

For both tasks, listeners following rules generated by PLLB (**Bottleneck**) outperform listeners trained from uninformed (**Baseline**) or misinformed (**Adversarial**) speakers (Figure 6). This holds true for all three different reward functions in BIRDS, as well as both training settings, with **Continual Learning** learning enabling **Bottleneck** to have an even stronger improvement over **Baseline** image descriptions by leveraging learned task experience. **Continual Learning** does not significantly help the **Adversarial** method, emphasizing that the linguistic rules $\mathcal{L}$ in **Bottleneck** do capture succesfull task strategies. However, **Continual Learning** is not required for **Bottleneck** to outperform **Baseline** image descriptions. The success of the **From Scratch** training setting can be interpreted as the speaker generating abstract rules to guide its own learning, leading to improved performance and showing evidence for a cognitive use of language.

Finally, we observe that the generated rules $\mathcal{L}$ encourage the speaker to reduce vague and general language, and describe properties relevant to the corresponding underlying reward function in BIRDS, although this might take several cycles to fully converge (see example rules in Appendix Section B.8). Furthermore, rules $\mathcal{L}$ become more complex over time, and across different trials we observe path dependency where different trials converge to different rules (e.g., $(x, y)$ vs. $x = 0.xx, y = 0.yy$ for coordinates in BUILDER). Overall, these rules help guide the speaker and listener agents in both tasks to iteratively improve image reconstruction over time, which can also be seen qualitatively in Appendix Figure 13.

## 7.3 PLLB can collaborate with human listeners

Do human listeners benefit from image descriptions provided by speakers following **Bottleneck** rules? We conduct a human subject study for the BUILDER task, where we replace the listener with study participants at the end of one iteration of PLLB, and evaluate how quickly they can reconstruct target images from a held-out test set. We recruit 20 crowdworkers on Prolific to reconstruct 5 images using descriptions from a speaker following a rule $\mathcal{L}$ generated with (**Bottleneck**), and 5 images using the original (**Baseline**) descriptions. We provided participants an interface that included a drawing canvas and buttons controlling shapes and colors (Section B.3), allowing us to use Levenshtein similarity between (sorted) user actions and the underlying action sequences for a target image as reward.

Participants using **Bottleneck** descriptions built target images faster and more accurately than participants using control descriptions (Wilcoxon signed-rank test, $p < 0.05$, Figure 6). In a post-study survey, 55% preferred **Bottleneck** descriptions, describing them as *"more direct and less ambiguous"*. 10% found both descriptions types equally easy to follow while remaining participants preferred **Baseline** descriptions because *they gave more flexibility*, leading to a more unstructured user study. However, we note that that user preference for customization does not correlate with any notion of improved performance in our original goal of image reconstruction.

## 7.4 PLLB collaborates with human speakers

Can humans benefit from rules generated by PLLB? We asked 12 Prolific crowdworkers to act as speakers and provide descriptions for our dataset of BIRDS images. Half of the participants were provided rules corresponding to one of the three reward functions ($\mathcal{L}_3$) while the other half were not. Figure 6 (bottom) shows that listeners instructed by human speakers provided with PLLB rules outperform those instructed by uninformed human speakers (**Baseline**) (Bonferroni-corrected t-tests $p < 0.05$ for **species** and **colors**).

The descriptions from users not provided a rule are more diverse and less focused e.g., *A barbed wire is an uncomfortable stop for a bird.* With PLLB rules on the other hand (e.g., *Describe the bird's coloration accurately*), they generate more specific descriptions: *Red and black bird on barbed wire. Bright red chest, red at top of head, black wings and beak* for the same target image (see other example rules in Appendix Table 4). PLLB agents can easily transmit what they learned from experience (the rule) to humans.

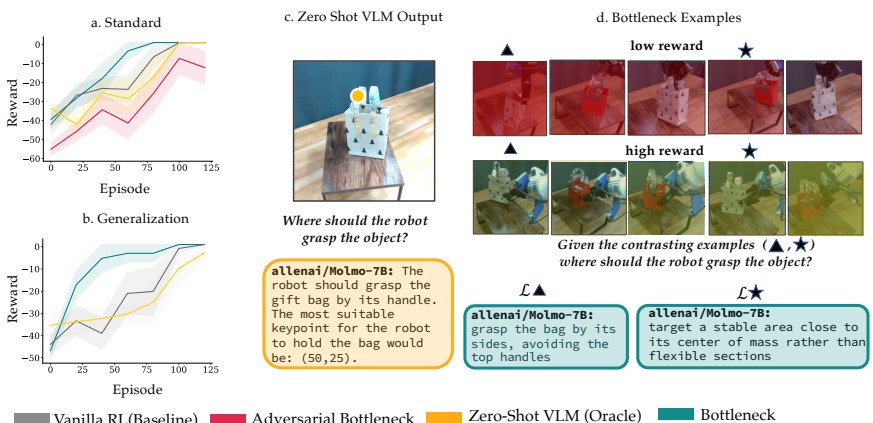

Figure 7: Results in GRASP. a) **Bottleneck** agents learn faster than the non-linguistic RL **Baseline agents**, as well as **Adversarial** and an **Oracle** zero-shot prompt of a VLM without any internal policy update. b) **Bottleneck** agents generalize faster than both **Baseline** (which over-relies on image details) and **Oracle** to unseen bag designs and locations, (c) shows the fixed linguistic rule d) Example grasps, shaded by reward.

## 8 Robot Grasp Planning

Can PLLB rules improve policy learning in embodied tasks with complex action spaces such as robot planning? Existing works on LM-based agents (e.g., ReAct (Yao et al., 2022) or Reflexion (Shinn et al., 2024)) and robot policies (e.g., Code as Policies) leverage pre-trained models for action generation, but cannot be applied in domains where a (non-LM-based) policy must be updated using real-world RL signals.

Consider training a robot to lift a paper bag. The most intuitive grasp would be at the bag's handles; indeed, a zero-shot **Oracle** prompt of the molmo-7b multi-modal language model outputs grasp keypoints on bag handles (Figure 7). However, if the bag is too heavy, a robot should adapt to less common grasps (e.g. by the side) to avoid tearing. Because VLMs are not grounded with actual physical quantities, such adaptation is impossible without leveraging an external reward (Gao et al., 2024). On the other hand, existing RL-based approaches for robotic manipulation are often inefficient (Kalashnikov et al., 2018). We show that PLLB rules can be used to efficiently learn non-intuitive robot grasping policies.

### 8.1 Task Overview

In the GRASP task, our agent is a Franka Emika Panda Arm robot that needs to lift and move a paper bag filled with unknown objects off of a shelf, as shown in Appendix Figure 12. Due to the complex action space, we follow Kalashnikov et al. (2018) and decompose the robot's policy into (i) perceiving the scene and identifying an appropriate grasp location, and (ii) planning a path to this location.

We begin with a *fixed* keypoint-conditioned path-planning policy which is pre-trained on a dataset of 50 trajectories with 3D-point cloud observations using Sphinx (Sundaresan et al., 2025). This model outputs a trajectory of 7-DoF end-effector poses (x/y/z/yaw/pitch/row/gripper) for the robot to execute given the keypoint. We then use PLLB to train (via RL) a grasp prediction policy that selects a grasp keypoint (a 2-D position) given an image of the environment. We define a successful grasp as one where the robot can lift the bag without it breaking, and so set our reward to be 0 for a successful grasp, -100 if the bag is not grasped at all, and use the normalized downward torque (e.g. -20Nm) applied to the robot's shoulder joint as a dense reward for unsuccessful grasps. Initially, the agent performs random grasps. After it observes one successful grasp, we run gen_rule with the open-source MOLMO-7B vision–language model with contrastive images of successful and unsuccessful grasp keypoints. This gives us a linguistic rule $\mathcal{L}$ (e.g. *grasp the left side of the bag*) that we can use to update the policy. We regularize the initial grasping policy toward the policy $\pi_{\mathcal{L}}$ induced from rule $\mathcal{L}$ by prompting molmo-7b while conditioning on $\mathcal{L}$, then instructing it to generate random keypoints. By executing Sphinx with these keypoints, the overall policy is updated to condition on $\mathcal{L}$.

We compare our **Bottleneck** method with an **Adversarial** version (swapping high and low reward grasps and an **Oracle** that always guides the policy with the rule generated from zero-shot prompting molmo-7b (no constrasting examples based on a reward). We call this Zero-Shot VLM an **Oracle** as it represents the VLM's privileged prior knowledge about intuitive grasping strategies, or knowledge a human expert would also possess. We also evaluate a **Vanilla RL** baseline where the policy only uses image observation data to predict a grasp keypoint based on task reward after execution, which helps us isolate PLLB's incremental value for keypoint selection.

### 8.2 PLLB can learn counter-intuitive policies

As shown in Figure 7, when averaged across 20 random seeds, our **Bottleneck** converges to higher reward compared to all other methods. Because the **Oracle** method suggests a rule encouraging the policy to predict key points near the bag's handles, it requires more episodes to learn that this behavior leads to lower reward. Qualitatively, we observe the generated rules converge towards guiding the robot to grasp at less common locations (e.g., *Grasp the bottom of the object*, see examples in Appendix Section B.8). This shows that PLLB can help improve over standard RL methods even for learning counter-intuitive policies.

### 8.3 PLLB improves few-shot generalization

To evaluate few0shot generalization, we create an environment with a new bag, location, and set of visual distractions (see Appendix 12). We then evaluate **Bottleneck**, **Zero-Shot VLM (Oracle)**, and **Vanilla RL (Baseline)** when intialized with the policies learned in the original environment, allowing for updates from the environment over time. As shown in Figure 7, the rules learned by **Bottleneck** outperform alternative methods for generalizing to this new environment. Because the **Vanilla RL** policy can only rely on image features, it is not able to leverage the more generalizable strategy PLLB rules capture and select grasp keypoints that hold semantic meaning (e.g. the bottom of the bag) captured by pre-trained LMs. Overall, the GRASP results show that PLLB can improve policy convergence even for the complex dynamics of embodied tasks.

## 9 Limitations & Discussion

PLLB helps agents becomes more interpretable by and inter-operable with humans. In our current implementation, `rule_gen` requires converting episodes into LM-compatible representations (text/images), which may limit use with long-horizon sensorimotor trajectories. Nevertheless, our Grasp task successfully uses pretrained robotic models, and advances in long-context modeling can address this limitation. A natural question is whether PLLB's gains arise specifically from language or from reward-aligned abstraction more generally. Comparing against non-linguistic bottlenecks (e.g., learned latent codes) is challenging: such representations require training across many task variations to become meaningful, whereas language provides pre-structured abstractions via pretrained LLMs that transfer to single tasks. Our **Adversarial** ablations across experiments confirm that rule content, not merely the presence of a bottleneck, drives improvements with PLLB.

Our human studies also demonstrate uniquely linguistic benefits (interpretability, human transfer) that would be unavailable with other latent representations (including code). However, rule generation can fail when the LM either abstracts too much (e.g., in BIRDS we once found the rule: *"Use descriptive language that conveys the mood, such as the serenity of a snowy day or the freedom of flight"*) or, more seriously, causes harm in safety-critical situations by creating a false trust in generated rules. We believe improvements in multi-modal models can help mitigate these issues (Appendix B.9), as well as additional human-in-the-loop verification steps of rules for high-stakes situations.

Our work opens up a set of interesting questions around designing intelligent sampling of contrastive episodes in `rule_gen`. For example, sampling episodes that are not outliers can help with handling stochastic environments, where high reward trajectories may not necessarily be optimal. Sampling of contrastive episodes can also adjust to known properties of how LMs attend to long context (i.e. more attention to episodes at the start and end). Another interesting direction for future work is applying PLLB to tasks with complex

reward functions that reflect hard-to-articulate human preferences, such as image captioning for accessibility (Nie et al., 2020) and personalizing language models (Li et al., 2023a). Although one concern is whether the linguistic rules from PLLB might exacerbate spurious reward signals, any reward misspecification becomes more detectable and diagnosable through interpretable rules, whereas black-box policies offer no such transparency.

Our experiments with PLLB focus on environments with tractable action spaces (e.g. discrete choices, text captions, keypoints for robotic grasping). In like open-ended text generation in math reasoning or code synthesis, this approach would require adaptation. Possible extensions include: (1) using the rule as a verifier or reward bonus rather than an action prior, (2) discretizing the action space into semantically meaningful categories that the LM can reason over, or (3) using the rule to filter or re-rank candidate actions from a base policy. Such adaptations would extend the cognitive and communicative benefits of language-guided learning to a broader class of sequential decision-making problems (Colas et al., 2022), and we leave exploration of these extensions to future work.

## 10 Acknowledgements

We would like to acknowledge support from NSF Awards #2006388, #2125511, #2238240 and #2212310, AFOSR YIP, ONR #N000142112298, and DARPA. Research was sponsored by the Department of the Air Force Artificial Intelligence Accelerator and was accomplished under Cooperative Agreement Number FA8750-19-2-1000. The views and conclusions contained in this document are those of the authors and should not be interpreted as representing the official policies, either expressed or implied, of the Department of the Air Force or the U.S. Government. The U.S. Government is authorized to reproduce and distribute reprints for Government purposes notwithstanding any copyright notation herein. Megha Srivastava was additionally supported by an IBM PhD Fellowship. Cédric Colas received funding from the European Union's Horizon 2020 research and innovation program under the Marie Skłodowska-Curie grant agreement No. 101065949.

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

# A   Appendix

# B   Appendix

## B.1   Hyperparameters

1. SAYSELECT: We use the same default parameters used in InstructRL (Hu & Sadigh, 2023), including setting the regularization strength $\lambda = 0.25$. Additionally, each time we invoke `gen_rule`, we create an ensemble of 3 rules, which we aggregate over when construct the probability distribution over actions during `update`.

2. MAZE: We use the InstructRL objective with a tabular Q-learning agent, but introduce a $\epsilon_{LM}$ parameter that controls whether the regularization strength $\lambda$ is 0 or 1 at each time-step in an episode. Although we did not observe a strong effect on modifying $\epsilon_{LM}$, we did not explore values larger than $\epsilon_{LM} = 0.4$ as that led to increased inference cost and experiment latency. Finally, each time we invoke `gen_rule` we create an ensemble of 4 rules, which we aggregate over when construct the probability distribution over actions during `update`.

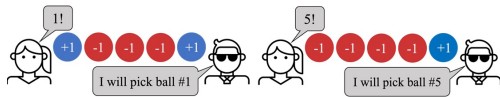

Figure 8: Overview of SaySelect game, reproduced from Hu & Sadigh (2023).

3. BUILDER: For the listener agent we finetune BART for a maximum of 100 epochs at each iteration of PLLB, employing early stopping on a held-out validation set and using the default arguments provided by the HuggingFace Transformers library (Lewis et al., 2019). Each time we invoke gen_rule, we same 3 rules and select the rule with the highest aggregate likelihood across all tokens. Likewise, for each image description (the speaker's action), we sample 3 possible descriptions under the given rule and select the one with the highest probability.

4. BIRDS: For the fine-tuned version of the listener agent, we finetune StableDiffusion for 1000 steps on a separate finetuning dataset. When generating images, we simply sample one image per description, using 10 inference steps and a guidance scale of 7.5.

5. GRASP: For grasp keypoint selection, we train an upper confidence bound agent on a neural contextual bandit using image features as context, with learning rate 0.01, with an $(100, 100)$ action space containing integer coordinates of the environment image. During update, we sample 20 coordinates following a given rule from gen_rule, which we use as the probability distribution over actions, setting $\lambda = 0.2$ as the regularization strength. To execute the grasp, we use a grasp model built by training Sphinx (Sundaresan et al., 2025) over 50 waypoint trajectories with learning rate 0.0001 for 100 epochs.

## B.2   SelectSay Overview

## B.3   Maze

See Figures 9 and 10.

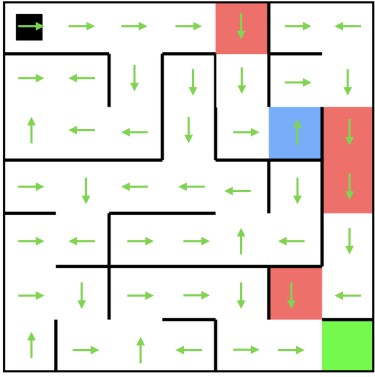

Figure 9: Visual aid provided to participants in the human subject study for the MAZE task.

## B.4   Dataset Details for Birds.

We consider images of the following bird species in the CUB-200-2011 dataset from Wah et al. (2011) for the BIRDS task: [Indigo Bunting, Cardinal, Yellow Breasted Chat, American Crow, Vermillion Flycatcher, California Gull, Blue Jay, Tropical Kingbird, White Pelican, Horned Puffin].

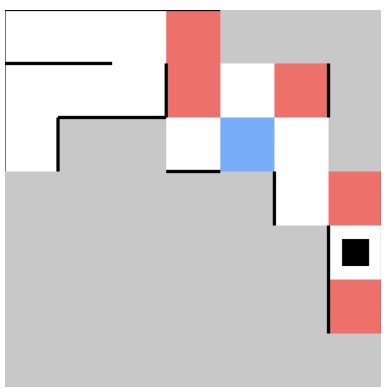

Figure 10: Interface used in the human subject study for the MAZE task. Cells not visited by the participants are hidden in gray.

**Welcome!**
Your goal is to follow the instruction on the top right to create an image using the buttons on the right. Each instruction belongs to either Type A or Type B - you should follow all instructions the same way, but we will ask you after the study to choose whether you found Type A instructions or Type B instructions easier to follow.
When you are done with one image, click the **Finish** button to proceed to the next one. You will create **10 images** in total. At the end, you will receive a link to a post-study survey to fill out.
Please complete the task in one sitting. Do your best to follow the instructions. We reserve the right to reject your submission if you create random or empty images.

**8/10**
**Instruction #8**
*(Type B)* Three red shapes in the top left quadrant, including two triangles and a square. One triangle located at (0.2, 0.73), another at (0.27, 0.72), and a third at (0.31, 0.84). Additionally, there is a red square positioned at (0.24, 0.25) and two more triangles, one at (0.23, 0.68) and the other at (0.21, 0.73).

| Red | Green | Blue | Magenta |
| Circle | Triangle | Square |
| Clear | Finish |

Figure 11: Interface used in the human subject study for the BUILDER task.

## B.5  Grasp

## B.6  Builder and Birds Iterations

## B.7  Full Prompts

We first provide the full prompts used for `gen_rule`. For space constraints, we do not include example `samples`, but note they follow the format shown in Figure 2.

1. SAYSELECT: You will be given a list of (OBSERVATION, ACTION, REWARD) examples collected from two agents learning to solve a task. Possible ACTIONS an agent can take are: 1, 2, 3, 4, 5, and quit. Each

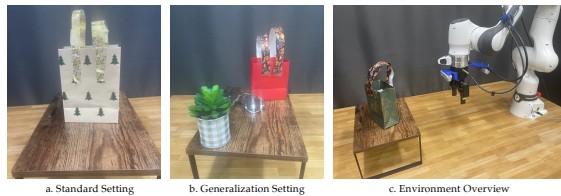

a. Standard Setting    b. Generalization Setting    c. Environment Overview

Figure 12: Task environment for GRASP, showing new location and distractors for the Generalization setting.

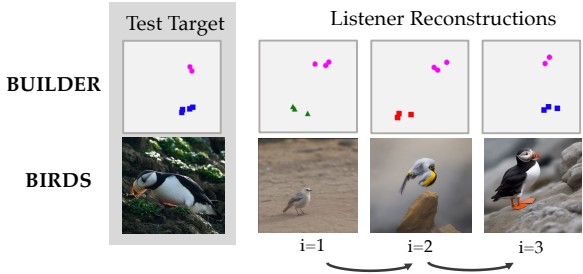

Figure 13: Both the BUILDER and BIRDS tasks consist of speaker and listener agents. At test time, the speaker needs to provide a language description to the listener that helps them recreate the image accurately. For both tasks, PLLB helps improve listener accuracy over time.

OBSERVATION describes the ordered sequence of actions that AGENT 1 picks, and each ACTION describes the ACTION that AGENT 2 picks based on the given OBSERVATION. The examples are separated into HIGH REWARD and LOW REWARD examples.+`[samples]`+Output a language rule that best summarizes the strategy AGENT 2 should follow to receive HIGH REWARD, not LOW REWARD, based on the examples. Start the instruction with the prefix 'I should'.

2. MAZE: You will be given a list of example (OBSERVATION, ACTION) trajectories collected from an AGENT learning to solve a maze. Each trajectory receives a REWARD. Possibles OBSERVATIONS an agent see are: WHITE, RED, BLUE Possible ACTIONS an agent can take are: NORTH, SOUTH, EAST, WEST. The examples are separated into HIGH REWARD and LOW REWARD examples + `[samples]` + Output a language rule that best summarizes the strategy the AGENT should follow when picking a sequence of ACTIONS to solve the maze and receive HIGH REWARD, not LOW REWARD, based on the examples. Start the instruction with the prefix 'I should'.

3. BUILDER: There are two agents. The goal of Agent 1 is to provide instructions to Agent 2 that helps Agent 2 to successfully recreate the image. You will be given a list of (ORIGINAL, AGENT 1 INSTRUCTION, RE-WARD) values where ORIGINAL is the original description of an image, INSTRUCTION is the instruction provided by Agent 1 to Agent 2, and REWARD is the reward Agent 2 receives when trying to re-create the image (higher is better). The examples are separated into HIGH REWARD and LOW REWARD examples. + `[samples]`+

Based on the examples above, output a list of 2 RULES for Agent 1 to follow when generating INSTRUC-TION in order to receive HIGH REWARD, instead of LOW REWARD. Write the rules after the prefix RULES:

4. BIRDS: The top row of three images have the following HIGH REWARD descriptions:+`high reward samples`+The bottom row of three images have the following LOW REWARD descriptions:+`low reward samples`+Provide a rule I should follow in order to provide image descriptions with HIGH REWARD, not LOW REWARD. Provide the rule after the prefix RULE:

5. GRASP: The top image shows a grasp keypoint with HIGH REWARD. The bottom image shows a grasp keypoint with LOW REWARD. Based on these images, provide a rule the robot should follow in order to select a grasp keypoint that results in HIGH REWARD, not LOW REWARD. Provide the rule after the prefix RULE:

We next provide the full prompts used in `update` for each task.

1. SAYSELECT: $[\mathcal{L}]$+Agent 1 selected `[observation]`. So I should select

2. MAZE: You are an agent solving a maze following a provided RULE. You will be given a list of PREVIOUS ACTIONS and the CURRENT OBSERVATION. Follow the RULE to select your NEXT ACTION (East, West, South, North):

   RULE: + $[\mathcal{L}]$+ PREVIOUS ACTIONS: + $[\tau_{1...t-1}]$ + CURRENT OBSERVATION: + `[observation]` +

   What is the NEXT ACTION you should take? Output one of (East, West, South, North) after the prefix NEXT ACTION:.

3. BUILDER: You will be given a DESCRIPTION of an image. Your goal is to use this description to provide a short INSTRUCTION to help someone else, who cannot see the image, accurately re-construct it. You will also be given a list of RULES you must follow when providing the instruction.

   DESCRIPTION: + `observation` +

   RULES: +$\mathcal{L}$ +

   Please provide a short instruction following the prefix INSTRUCTION:

4. BIRDS: Provide a one-sentence description of this image, using the following RULES:+$\mathcal{L}$

5. GRASP: Provide a keypoint in the image where the robot should grasp the object, following the RULE: $\mathcal{L}$.

## B.8 Examples of Generated Rules for All Environments

See Tables 1, 2, 3, 4, and 5.

| Setting | $\mathcal{L}_1$ | $\mathcal{L}_2$ | $\mathcal{L}_5$ |
|---|---|---|---|
| Standard | I should follow the strategy of choosing action 4 whenever the observation is 2, 3, 4, or 5. | I should follow the strategy of choosing actions based on the observation provided by AGENT 1. If the observation is 1, I should choose action 1. If the observation is 2, I should choose action 3. If the observation is 3, I should choose action 3. If the observation is 4, I should choose action 4. If the observation is 5, I should choose action 5. | I should follow the strategy of choosing the same action as AGENT 1 for observations 1, 2, 3, 4, and 5. |
| Fixed Speaker | I should choose action 1 when observation is 1 or 2 or 4 or 5. I should choose action 2 when observation is 3. | I should choose action 1 when observation is 1, 2, 3 or 5. I should choose action 2 when observation is 4. | I should choose action 1 when observation is 1. I should choose action 4 when observation is 2. I should choose action 5 when observation is 3. I should choose action 2 when observation is 4. I should choose action 3 when observation is 5. |

Table 1: Example $\mathcal{L}$ rules generated for the SAYSELECT environment, for the Standard setting (both Listener and Speaker agents are RL agents trained from random initialization) and a Fixed Speaker agent.

## B.9 Effect of Model on Generated Rules

We additionally compare the effect of different language models, in particularly those of different sizes, on our results. We did not observe a significantly strong quantitative difference in performance when `gen_rule` is instantiated with models of different sizes (e.g. llama-2-70b-chat vs. llama-2-13b-chat). However, we did notice interesting qualitative differences across samples that are likely due to the additional fine-tuning step using reinforcmenet learning from human feedback (RLHF, see https://llama.meta.com/llama2/ for more information). We describe these differences per environment below.

1. SAYSELECT: Because of the simplicity of this environment, it is possible for a rule $\mathcal{L}$ to summarize the full optimal human-interpretable policy as a sequence of if-statements (e.g. *"If the current state is '1', I should take action 1."*) - we observed that smaller language models (e.g. llama-2-7b-chat and llama-2-13b-chat) always did this, while larger models (e.g. llama-2-70b-chat and mistral-8x-7b-instruct) were better able to generalize and use more efficient language, such as *"I should take the same action as the observation"*. The smaller models also often included superfluous language, such as *"I should always take the action that leads to the highest reward"*.

2. MAZE: The majority of generated rules captured the underlying color semantics of the maze, enabling generalization. However, smaller model sizes (e.g. llama-2-7b-chat) resulted in more superfluous language (e.g. *"I should always prioritize taking actions that lead to the most recent reward, and avoid taking actions that lead to low reward."*), and the generated rules focused more on the first few actions the agent should take, which may not always generalize.

3. BUILDER: While we do not observe any model-specific differences, there exists variation across samples in the type of formatting and syntax generated rules encourage (e.g. provide coordinates *" using the format (x, y)"* vs. *" using the format "x=0.XX, y=0.YY""*), leading to agents converging to different descriptions. Furthermore, some rules encourage list formats in image descriptions (e.g. *1.Draw a green dot at (0.72, 0.21). 2. Draw a green dot at (0.73, 0.72).*) while other rules encouraged clustering of identical shapes (e.g. *Draw two green dots at (0.72, 0.21) and (0.73, 0.72).*)

| Size | $\mathcal{L}_1$ | $\mathcal{L}_2$ | $\mathcal{L}_5$ | $\mathcal{L}_8$ |
|---|---|---|---|---|
| 7x7 | I should start by moving north, then continue moving north until I reach a wall, then turn east and move until I reach a wall, then turn south and move until I reach a wall, then turn west and move until I reach the end of the maze. | I should prioritize moving North, East, and South over moving West, as the HIGH REWARD EXAMPLES show a higher frequency of these actions. 2. I should avoid moving West unnecessarily, as it is rarely seen in the HIGH REWARD EXAMPLES and often leads to LOW REWARD EXAMPLES | I should take the EAST action when I see WHITE, and then repeat the EAST action until I see RED, then take the SOUTH action, and then repeat the EAST action until I see BLUE, then take the NORTH action, then take the EAST action until I see RED again, then take the SOUTH action, and finally take the WEST action. | I should take the EAST action when I observe WHITE, and keep taking EAST actions until I observe RED, then take the SOUTH action. After taking the SOUTH action, I should take the EAST action again until I observe BLUE, then take the NORTH action. After taking the NORTH action, I should take the EAST action until I observe RED again, then take the SOUTH action. |

Table 2: Example $\mathcal{L}$ rules generated for the MAZE environment for the 7x7 maze size.

| Setting | $\mathcal{L}_1$ | $\mathcal{L}_2$ | $\mathcal{L}_3$ |
|---|---|---|---|
| Re-Initialization | 1. Be specific with location details: Agent 1 should provide detailed location information for each element in the image, such as x and y coordinates. 2. Use descriptive language for elements, such as "red dot" or "green triangle". | 1. Use specific coordinates when instructing Agent 2 to draw shapes. 2. Use descriptive language to specify the color and shape of each element. For example, "a green triangle" instead of "a green thing". | 1. Be specific with location coordinates: provide specific coordinates for the location of each shape, using the format x=0.XX, y=0.YY. 2. Use descriptive shape names: Instead of using generic terms like "dot" or "square," use more descriptive names that indicate the shape's color and size, such as "green triangle" or "red square." |
| Continual Training | 1. Be specific and detailed in your instructions. High reward examples have specific coordinates and shapes, while low reward examples have more general descriptions. 2. Use a consistent format for your instructions. High reward examples have a consistent format for listing coordinates and shapes, while low reward examples have a more free-form format. | 1. Provide explicit coordinates for each element in the image, using the format (x, y). 2. Use specific colors when referring to elements in the image, such as "red", "green", or "blue". Avoid using vague terms like "colored" or "shaded". | 1. Use a consistent format for describing shapes, such as always listing the x-coordinate first, followed by the y-coordinate. For example, instead of "one green square at the point x=0.53, y=0.24", use "one green square at (0.53, 0.24)". 2. Avoid using vague terms like "various shades of green". Instead, use specific colors, such as "green" or "blue". Additionally, use specific shapes, such as "square" or "triangle", rather than vague terms like "rectangle". |

Table 3: Example $\mathcal{L}$ rules generated for the BUILDER environment.

4. BIRDS: Rules demonstrated more reward-specificity (i.e. specific to background, color, or species rewards) when generated with larger VLMs (e.g. llava-13b) versus smaller models (e.g. llava-v1.6-vicuna-7b), with the latter primarily proposing rules that encouraged more detailed descriptions (e.g. *"Avoid using vague or general terms"*).

5. GRASP: We found that most VLMs other than molmo-7B performed poorly at recognizing keypoints, and instead relied heavily on task information. For example, one rule generated by llava-13b was *"A robot should grasp a keypoint that is visible and not obstructed by other objects."* We believe the superior performance of the molmo series of models are due to specifically training on the PixMo dataset with 2D-points (Allen Institute for AI).

| Reward | $\mathcal{L}_1$ | $\mathcal{L}_2$ | $\mathcal{L}_3$ |
|---|---|---|---|
| **color** | Describe the bird's color, species, and any distinctive markings or patterns. | Describe the bird's coloration accurately. | Describe the bird's coloration accurately. |
| **background** | Include details about the bird's surroundings, such as the type of branch or post it is on, and any additional elements in the background. | Include the bird's action (perched, flying, standing) and its location (on a branch, railing, pole, etc.) | Describe the bird's action (flying, perching, standing) and the environment it is in (sky, tree, water). |
| **species** | Describe the bird's color, markings, and any distinctive features. | Describe the subject's unique features, such as coloration, beak shape, or other distinguishing characteristics. | Include specific details about the bird's appearance, such as the color of its feathers, beak, or eyes, and any distinctive markings or patterns. |

Table 4: Example $\mathcal{L}$ rules generated for the BIRDS environment demonstrate reward-specificity over time.

| Reward | $\mathcal{L}_{10}$ | $\mathcal{L}_{30}$ | $\mathcal{L}_{50}$ |
|---|---|---|---|
| **Shoulder Torque** | Select keypoints that are away from sharp edges or corners of the object to avoid potential damage and improve stability during grasping. | Prioritize grasp keypoints that are closer to the center of the object for more stable and precise pickup. | Prioritize grasping at the middle of the object's surface rather than the top edges or handles to ensure a stable grip. |

Table 5: Example $\mathcal{L}$ rules generated for the GRASP environment.

## B.10 Sensitivity Analysis of Rule Generation

We next compare the sensitivity of PLLB to the implementation of `gen_rule`, specifically variations in the prompt and temperature for sampling. We present results for SAYSELECT, as its discrete action space and deterministic dynamics provide the clearest signal for detecting sensitivity effects.

### B.10.1 Prompt Variations

For SAYSELECT, we evaluate prompt sensitivity in `gen_rule` by comparing the following variations with the **Adversarial** baseline described in 5.:

- (**Original**) You will be given a list of (OBSERVATION, ACTION, REWARD) examples collected from two agents learning to solve a task. Possible ACTIONS an agent can take are: 1, 2, 3, 4, 5, and quit. Each OBSERVATION describes the ordered sequence of actions that AGENT 1 picks, and each ACTION describes the ACTION that AGENT 2 picks based on the given OBSERVATION. The examples are separated into HIGH REWARD and LOW REWARD examples.+[samples]+Output a language rule that best summarizes the strategy AGENT 2 should follow to receive HIGH REWARD, not LOW REWARD, based on the examples. Start the instruction with the prefix 'I should'.

- (**No Format Instruction**) You will be given a list of (OBSERVATION, ACTION, REWARD) examples collected from two agents learning to solve a task. Possible ACTIONS an agent can take are: 1, 2, 3, 4, 5, and quit. Each OBSERVATION describes the ordered sequence of actions that AGENT 1 picks, and each ACTION describes the ACTION that AGENT 2 picks based on the given OBSERVATION. The examples are separated into HIGH REWARD and LOW REWARD examples.+[samples]+Output a language rule that best summarizes the strategy AGENT 2 should follow to receive HIGH REWARD, not LOW REWARD, based on the examples.

- (**Low Context**) You will be given a list of (OBSERVATION, ACTION, REWARD) examples collected from two agents learning to solve a task. Output a language rule that best summarizes the strategy AGENT 2 should follow to receive HIGH REWARD, not LOW REWARD, based on the examples. Start the instruction with the prefix 'I should'.

- (**Rephrase**) You've been given a list of (OBSERVATION, ACTION, REWARD) triples from two agents learning to solve a task. Possible ACTIONS each agent might take are: 1, 2, 3, 4, 5, and quit.

Each OBSERVATION refers to the ordered sequence of actions that AGENT 1 selects, and each ACTION refers to the ACTION that AGENT 2 selects based on the seen OBSERVATION. The examples are divided into HIGH REWARD and LOW REWARD examples.+[samples]+Describe the strategy AGENT 2 uses in HIGH REWARD examples that differs from LOW REWARD examples. Start it with 'I should'.

Table 6 and 7 show that PLLB is mostly robust to variations in prompt and always outperforms the **Adversarial** baseline, showing that PLLB indeed places higher weight on the content of the contrasting episodes rather than relying on any specific syntax for rule generation.

### B.10.2 Temperature Variation

For SAYSELECT, our default temperature for sampling with `gen_rule` is 0.5. We compare with temperatures 0.1 (low diversity) and 0.9 (high diversity), and find only slight decrease in reward at the higher temperature of 0.9 (see Table 6). Importantly, across all temperatures the average reward and human-interpretability scores are stronger than baselines, indicating robustness towards sampling temperature.

| Episode | 500 | 1000 | 6000 |
|---|---|---|---|
| Reward (**Original, temp=0.5**) | $0.63 \pm 0.1$ | $0.8 \pm 0.1$ | $0.96 \pm 0.03$ |
| Reward (**Original, temp=0.9**) | $0.47 \pm 0.2$ | $0.68 \pm 0.1$ | $0.96 \pm 0.05$ |
| Reward (**Original, temp=0.1**) | $0.66 \pm 0.1$ | $0.70 \pm 0.09$ | $0.96 \pm 0.02$ |
| Reward (**No Format Instruction**) | $0.51 \pm 0.2$ | $0.77 \pm 0.05$ | $0.96 \pm 0.02$ |
| Reward (**Low Context**) | $0.56 \pm 0.2$ | $0.73 \pm 0.1$ | $0.96 \pm 0.02$ |
| Reward (**Rephrase**) | $0.64 \pm 0.1$ | $0.81 \pm 0.2$ | $0.96 \pm 0.02$ |
| Reward (**Adversarial**) | $0.21 \pm 0.1$ | $0.45 \pm 0.03$ | $0.8 \pm 0.05$ |

Table 6: Reward across ablations to probe sensitivity of rule generation to prompt variation and temperature. Results are averages across 5 trials.

| Episode | 500 | 1000 | 6000 |
|---|---|---|---|
| Interpretability (**Original, temp=0.5**) | $0.77 \pm 0.03$ | $0.87 \pm 0.04$ | $0.97 \pm 0.02$ |
| Interpretability (**Original, temp=0.9**) | $0.72 \pm 0.05$ | $0.80 \pm 0.05$ | $0.93 \pm 0.03$ |
| Interpretability (**Original, temp=0.1**) | $0.87 \pm 0.04$ | $0.90 \pm 0.1$ | $0.95 \pm 0.02$ |
| Interpretability (**No Format Instruction**) | $0.67 \pm 0.06$ | $0.91 \pm 0.03$ | $0.94 \pm 0.02$ |
| Interpretability (**Low Context**) | $0.77 \pm 0.04$ | $0.84 \pm 0.05$ | $0.97 \pm 0.03$ |
| Interpretability (**Rephrase**) | $0.8 \pm 0.04$ | $0.97 \pm 0.04$ | $0.97 \pm 0.02$ |
| Interpretability (**Adversarial**) | $0.55 \pm 0.03$ | $0.57 \pm 0.03$ | $0.65 \pm 0.06$ |

Table 7: Human interpretability across ablations to probe sensitivity of rule generation to prompt variation and temperature. Results are averages across 5 trials.

### B.11 Non-Contrastive Variant

A core feature of PLLB is using contrastive episodes (examples of both low and high reward episodes) to `gen_rule`. Here we report a non-contrastive ablation for the SAYSELECT (RL) and BIRDS (diffusion-model based communication game) environments, where we only use high reward examples in the prompt to `gen_rule`. Our results show that contrastive episodes are important for strong performance. Concretely, for SAYSELECT, the non-contrastive ablation decreases both task reward and human-interpretability of the RL policy, across different total number of episodes seen (see Table 8). Likewise, for the BIRDS task, the non-contrastive ablation results in lower reward than PLLB, although still leads to a non-trivial improvement over the Baseline and Adversarial methods (see Table 9). Upon a closer look, we find that without contrasting episodes (i.e. negative episodes that highlight what behaviors are shared between high and low rewarding strategies), PLLB rules often capture spurious features of the tasks, such as *" Include details about the*

*bird's actions. This helps in creating a more vivid and engaging description."*. This subsequently results in highly-specific captions such as a bird *'possibly displaying a territorial or mating display'* or *'looking at the camera'*, neither of which are relevant for any of our reward functions. Overall, these results highlight the importance of using contrasting episodes to improve PLLB rule equality.

| Episode | 500 | 1000 | 6000 |
|---|---|---|---|
| Reward (**Original**) | $0.63 \pm 0.1$ | $0.8 \pm 0.1$ | $0.96 \pm 0.03$ |
| Reward (**Non-contrastive**) | $0.53 \pm 0.02$ | $0.7 \pm 0.2$ | $0.91 \pm 0.03$ |
| Reward (**Adversarial**) | $0.21 \pm 0.1$ | $0.45 \pm 0.03$ | $0.8 \pm 0.05$ |
| | | | |
| Interpretability (**Original**) | $0.77 \pm 0.03$ | $0.87 \pm 0.04$ | $0.97 \pm 0.02$ |
| Interpretability (**Non-contrastive**) | $0.75 \pm 0.02$ | $0.8 \pm 0.06$ | $0.88 \pm 0.02$ |
| Interpretability (**Adversarial**) | $0.55 \pm 0.03$ | $0.57 \pm 0.03$ | $0.65 \pm 0.06$ |

Table 8: Reward and human-interpretability scores for contrastive and non-contrastive versions of PLLB in SAYSELECT. Results are averages across 5 trials.

| (a) Color | | (b) Background | | (c) Species | |
|---|---|---|---|---|---|
| Method | Score | Method | Score | Method | Score |
| **Baseline** | $1.69 \pm 0.5$ | **Baseline** | $1.27 \pm 0.5$ | **Baseline** | $1.59 \pm 0.5$ |
| **Bottleneck** | $2.93 \pm 0.4$ | **Bottleneck** | $2.30 \pm 0.3$ | **Bottleneck** | $2.69 \pm 0.5$ |
| **Non-contrastive** | $2.64 \pm 0.3$ | **Non-contrastive** | $1.78 \pm 0.4$ | **Non-contrastive** | $2.51 \pm 0.5$ |
| **Adversarial** | $1.43 \pm 0.5$ | **Adversarial** | $1.01 \pm 0.4$ | **Adversarial** | $1.21 \pm 0.4$ |

Table 9: Noncontrastive ablation in comparison with main results for BIRDS task across 3 reward functions. Results are averaged across 5 trials.

