# OpenReview forum: "Policy Learning with a Language Bottleneck"
_TMLR — Accepted by TMLR_

### Review · Reviewer_4Yfh · 2025-11-20

**Summary Of Contributions:**

This paper introduces Policy Learning with a Language Bottleneck (PLLB), a framework that incorporates a language-based feedback loop for policy improvement. The core idea is to alternate between two steps: (1) `gen_rule`, where a language model (LM) is prompted with contrastive (high vs. low reward) episodes to generate a linguistic rule summarizing successful strategies, and (2) `update`, where the agent's policy is updated, regularized by the generated rule. The authors evaluate PLLB across five diverse tasks (SaySelect, Maze, Builder, Birds, and Robot Grasp Planning) to demonstrate its benefits on interpretability and generalization, etc.

**Audience:**

Yes

**Audience Explanation:**

Yes, this paper sits at the intersection of several high-interest areas for the TMLR audience and would be of significant interest to a broad segment of its readers. To name a few:

* Reinforcement learning researchers: They would be interested in a novel training paradigm that imbues RL agents with interpretability and generalization without sacrificing performance.
* Natural language processing researchers: The paper presents an innovative application of LMs not just as reasoners or policy executors, but as tools for rule abstraction and distillation.

**Broader Impact Concerns:**

One potential concern is the misleading or incorrect rules. If a PLLB agent generates a flawed rule that is nonetheless convincing to a human, it could lead to the human making errors. The "Adversarial" ablation shows that bad rules can hinder learning, but the risk of a plausible-but-wrong rule causing harm in a safety-critical domain (e.g., a self-driving car's rule being misinterpreted by a engineer) is a non-trivial risk that should be acknowledged. Future work should include safeguards, such as human-in-the-loop verification for high-stakes rules.

**Claims And Evidence:**

Yes

**Claims Explanation:**

Yes. Here are a few claims by the paper and their evidences:

Claim 1. PLLB learns more interpretable and generalizable behaviors.

Evidence: Interpretability (SaySelect): PLLB agents converge to the human-preferred policy (e.g., "choose the same number") without external supervision, while a standard RL baseline does not. Generalization (Maze): PLLB agents transfer knowledge to new mazes with the same color semantics much faster than baselines. The "Adaptation" experiment further shows they can unlearn an old rule and learn a new one more effectively than a linear function approximator, which overfits.

---

Claim 2. PLLB enables more effective human-AI coordination (inter-operability).

Evidence: This is validated through three separate human subject studies: (a) Maze: Humans given PLLB-generated linguistic rules solved mazes faster than those given visual aids or no aid, and found the rules more useful. (b) Builder: Human "listeners" reconstructed images more accurately and faster using descriptions from PLLB-guided "speakers." (c) Birds: Human "speakers" provided with PLLB rules generated descriptions that led to higher-reward image reconstructions than uninformed humans.

---

In all experiments, the authors use an "Adversarial" ablation (using corrupted rules) as a control group. It shows that the quality of the generated rule is crucial, not just the presence of any linguistic signal. This rules out the possibility that the LM is merely providing a useful inductive bias regardless of content.

**Requested Changes:**

To further improve its clarity and rigor, the following changes are requested:

* Clarify the `update` step. From 3.2 I understand that the update step involves a Q-learning objective. How is it implemented for LLM? And how the regularization term works in this context? More rigorous technical details are needed to improve the clarity of the method.

* Scalability. Can the method be extended to domains such as math reasoning where the action space is not limited to a few choices?  Please illustrate how this method can be extended and whether the Q-learning objective is still applicable.

---

> ### Author Response · Authors · 2025-12-21
> **Rebuttal**
>
> Thank you for your helpful feedback! We address each point under the Requested Changes below:
>
> 1. Update Step: *We have updated Section 3.2 to be more clear that the update step depends on the underlying agent.* The update step only involves a Q-learning objective when the agent’s policy is trained via Q-learning (and similarly, a PPO-style agent can use the InstructPPO objective from Hu & Sadigh), and we perform a softmax over the logits for all possible actions. For policies purely implemented with LLMs/VLMs (including for continuous action spaces such as Grasp), update can simply be executed by adding to the prompt.
>
> 2. Scalability: The primary limitation with scale of PLLB is the context length required to fit contrastive episodes with high/low reward. For sufficiently complex math reasoning problems, there may not be sufficient context length to fit entire trajectories. However, a large, non-discrete action space is not an issue for PLLB, as our Grasp experiment shows PLLB can be used for continuous action spaces (keypoint pixels). We note that the Q-learning objective is only relevant for tasks where the policy is implemented as a Q-learning agent; if the policy is based on an LM, then rules can be incorporated via prompt conditioning as shown in Section 6. *We have updated Section 3.2 and Section 9 Limitations to address this, including mentioning how to adapt PLLB to domains like math reasoning.*
>
> 3. Broader Impact: We acknowledge the reviewer’s concerns about flawed rule generation, and have *updated Section 9 Limitations to reflect this.*
>
> Thank you for your feedback that has helped us to improve the paper.

---

### Review · Reviewer_oXZv · 2025-11-23

**Summary Of Contributions:**

This paper proposes a method called Policy Learning with a Language Bottleneck (PLLB) which combines reinforcement learning with LLM-based control. The authors argue that this combination is better than either technique alone. RL can struggle to generalize to new environments and often produces convoluted, uninterpretable policies. LLM-based control can generalize better and is more interpretable, but struggles to learn policies that are difficult to express with language. PLLB alternates between prompting an LLM to propose rules that explain the difference between high- and low-reward trajectories, and then updating the current policy with Q-learning steered towards actions proposed by the generated rule. The authors test PLLB across five environments: a collaborative tabular environment, a single-agent tabular environment, two visual environments, and a robotics environment. They find that PLLB generally produces more interpretable and generalizable policies compared to RL, while succeeding in some environments where LLM-based control struggles.

**Audience:**

Yes

**Audience Explanation:**

There has been quite a bit of research interest in combining LLMs with classical control and RL techniques in recent years, so it seems like there would definitely be an audience for this paper.

**Broader Impact Concerns:**

No concerns.

**Claims And Evidence:**

Yes

**Claims Explanation:**

The experiments in the paper are quite comprehensive and mostly back up the authors' claims that PLLB produces policies that are more performant, generalizable, interpretable, and inter-operable. I think the shakiest claim is "generalization," which seems to be evaluated in the Maze and Grasp environments. In the RL literature, generalization often refers to the ability to perform well in new environments *with no additional training* [1, 2]. However, the experiments in this paper evaluate "generalization" performance after training in the new environment. I think a more accurate description for this would be "adaptation" or at least "few-shot generalization."

[1] Cobbe et al. Leveraging Procedural Generation to Benchmark Reinforcement Learning (evaluates generalization to new environments with no additional learning)

[2] Fu et al. Learning Robust Rewards with Adversarial Inverse Reinforcement Learning (refers to GAIL failing to generalize in terms of zero-shot performance in a different environment)

**Requested Changes:**

My main request is to rework the claim of "generalization" to be in line with other deep RL papers.

Additionally, it seems like an undiscussed limitation of PLLB is its extension to stochastic environments. In general, methods that try to construct a policy by contrasting low- and high-reward trajectories can lead to overoptimism in stochastic environments. For example, if an agent has a choice between taking action A for a sure $10 or action B for a 10% chance of $20, the optimal action is to take action A. However, the trajectories with the highest rewards will all involve taking action B. It's unclear if PLLB will work in such a setting, and it seems worth at least mentioning that stochastic environments could be a challenge.

Besides the above, I think there are a few areas where the clarity of the paper could be improved:
 * Figure 2 shows an overview of how PLLB learns in each environment, but it doesn't actually describe the environments, making it quite confusing before reading through the rest of the paper. It would be helpful to have some visual representation of the environment, what the actions are, what the objective is, etc. in order to understand what's going on. At first it's also hard to understand what the top row of the figure is (the boxes that say "Output a rule that best summarizes the strategy I should follow", etc.).
 * The "adversarial bottleneck" condition is briefly described in Section 4 but there is little explanation for why it is included in the experiments. I think I understand that the idea is to make sure that PLLB is actually using the language guidance, since if it was ignoring the language rules then the adversarial setting wouldn't hurt performance. It would be helpful to make the purpose of this condition more clear in Section 4.
 * I also found the "oracle" label to be unclear in the Grasp experiments. I think it makes sense for SaySelect, because in this case the oracle is given access to the rule it should follow. However, in Grasp, the paper says the oracle “always guides the policy with the rule generated from zero-shot prompting molmo-7b." This doesn't really seem like an oracle setting since it's not given additional information that the PLLB policy can't access. Also, how exactly does the rule guide the policy? Via InstructRL? It would be helpful to have more details.
 * Small typo: there is a missing space after LinearQ in Figure 4's caption.

---

> ### Author Response · Authors · 2025-12-21
> **Rebuttal**
>
> Thank you for your thoughtful feedback! We address each point under the Requested Changes below:
>
> 1. Generalization Claims: **We have updated the Introduction, Section 6 and 8** to use the phrase “few-shot generalization”, noting that the term “adaptation” was used in Maze to describe adapting to new environments with completely different structures (where a rule would not generalize).
> 2. Stochastic Environments: **We have updated the Limitations section** to address this, noting that one potential way to mitigate this concern could be more intelligent sample of contrasting low/high reward trajectories.
> 3. Figure 2: **We have updated Figure 2** to make the environments clearer visually, as well as edited the caption to indicate the gray boxes contain LM prompts.
> 4. Adversarial Bottleneck: The intention of the adversarial bottleneck is to verify that language based on the agent’s own experience (via contrasting episode pairs) is the cause of PLLB’s performance, versus any language rule simply providing appropriate conditioning. **We have updated Section 4 to better clarify this.**
> 5. Grasp Experiment:  The reviewer is correct that the Oracle here does not access privileged information – we call it Oracle to reflect the privileged information a VLM has about intuitive grasping strategies (similar to how we refer to InstructRL as an Oracle in the Fixed Speaker experiment in Section 5) and our goal is to test whether this is sufficient. Our results show that reward from the environment is necessary to improve the policy. **We have updated Section 8.1 to improve clarity on this and include more details.**
> 6. Typo: We have fixed the typo
>
> Thank you again for your feedback that has helped us to improve the paper.

---

### Review · Reviewer_h2d6 · 2025-12-08

**Summary Of Contributions:**

The authors provide a framework for learning RL agents to generate linguistic rules that capture the high-level strategies underlying rewarding behaviors, and use these rules to learn a policy that solves the task

The proposed procedure involves two steps:
- **Rule generation:** a LM is used to generate rules that explain a set of observed trajectories. This LM is shown the task description and contrastive examples, specifically, the the top N and bottom N rewarding trajectories from the collected data. The LM is then asked to produce a concise rule of the form “I should …” that explains what differentiates successful from unsuccessful behavior.
- **Policy Update:**  The agent then uses the rule to shape behavior in a domain-appropriate way: in the RL settings, the rule induces a rule-conditional action prior from the LM (via a softmax over logits) that regularizes Q-learning; in the non-RL collaborative reconstruction settings, the rule is iteratively appended to the speaker’s prompt to regenerate more reward-aligned instructions and improve downstream listener performance over successive rounds.

By inserting a self-generated language bottleneck between experience and learning, the authors argue that PLLB improves performance across diverse settings, promotes the discovery of reusable high-level abstractions, and yields rules that are directly useful for human-agent coordination.

**Audience:**

Yes

**Audience Explanation:**

I think the paper will be interesting to some communities in TMLR. It explores the intersection between iterative learning loops in sequential decision making and the use of self-generated language as an intermediate representation within that loop. The framing of language as a bottleneck that can shape subsequent learning, coordination, and generalization is likely to resonate with readers interested in representation learning, agent communication, and the role of LMs in RL.

**Broader Impact Concerns:**

No concerns

**Claims And Evidence:**

Yes

**Claims Explanation:**

The paper shows the following core contributions/claims:
- Language can act as a useful bottleneck even when the optimal policy is only partially expressible in text → The strongest evidence in support of this claim is the failure of the pure LM based policy that does no Q value updates on the Maze task.
- PLLB can steer coordination toward human-preferred conventions, increasing interpretability comparable to an instruction-oracle baseline without external instruction → This claim is supported in the SaySelect task, where PLLB and InstructRL both converge to the human-interpretable policy, while TabularQ does not, indicating that the self-generated rule plays the same steering role as the oracle instruction.
- PLLB yields better generalization and adaptation in structured environments relative to tabular and linear baselines, with rules converging toward the latent color-to-action semantics → This claim is supported in the Maze experiments, where the authors compare PLLB to tabular Q that has no notion of color, and linear Q which uses a feature related to color. When the authors keep the semantics the same, but change the optimal policy, PLLB is the best. When the color semantics are changed, PLLB adapts faster to the new rule. Post-hoc analysis of the rules validates that rules converge to color-to-action semantics
- Providing users with PLLB-generated rules improves performance and perceived usefulness, showing that self-generated agent rules have communicative value → This claim is supported by the human-based study on the Maze task where human subjects learn to solve the maze in fewer steps when given PLLB rules as compared to no help (control) and visual aid (visual baseline)
- The observed gains arise from reward-aligned rule extraction rather than adding language per se → This claim is argued for mostly through the adversarial baseline for the experiments
- Self-generated rules improve multi-agent coordination quality. → This claim is primarily shown in the non-RL settings, specifically Builder and Birds. Here, the learning is not Q-learning, but rather the  learned rule is iteratively appended to the speaker’s prompt, re-generating descriptions, and re-training or updating the listener. In such cases, PLLB rules help the speaker produce more usable and less ambiguous instructions than baseline prompting for both synthetic and neutral images.

The evidence, as it stands, supports the claims to a great extent. However, it can be strengthened by addressing the changes requested.

**Requested Changes:**

- **Pitch:** Currently, I am currently struggling to identify a crisp core novelty that unifies the broad PLLB instantiations. Because the paper spans regularized Q-learning, instruction-augmented image reconstruction, and a bandit-style learner in Grasp, I suggest the authors articulate the common mechanism that is invariant across these settings more clearly. It would significantly strengthen the central pitch.
- **The necessity of language as a bottleneck:**  The results currently show that a bottleneck derived from reward contrast can help learning and coordination. In this work, this bottleneck takes hte form of language rules. However, they do not yet convincingly show that language is the essential form this bottleneck must take. I would request a baseline that uses a similarly sized non-linguistic bottleneck derived from the same contrastive data (e.g.; a learned contrastive classifier or compact latent code). This would directly test whether the gains are language-specific or reflect a broader reward-aligned abstraction effect.
- **The link between the PLLB rules and rewards:** The paper’s framing is that gains arise from reward-aligned rule extraction rather than adding language per se. I think this is the right hypothesis to test, and the adversarial bottleneck is a good start. However, given that reward design heavily shapes what the rules should look like, I worry that PLLB may also inherit and potentially amplify reward mis-specification. The paper notes that some participants preferred baseline descriptions in communication tasks because they offered more flexibility, which may indicate the reward function does not fully reflect user preferences. I would like the authors to either argue more carefully why this does not undermine the reward-alignment claim, or provide additional analyses demonstrating how rule quality tracks meaningful reward structure without simply reinforcing reward artifacts.
- **Rule quality and robustness:** Rule generation relies on prompting an LM with contrastive trajectories. I would like to see sensitivity analyses w.r.t. the number of positive/negative examples, the sampling choice, and modest prompt variations. This would help establish how robust the method is to design decisions in the rule-generation pipeline.
- **Contrastive Ablation:** Reward contrast appears to be the centerpiece of PLLB’s rule quality. I would request an explicit ablation comparing contrastive rule generation to a non-contrastive variant (e.g., high-reward-only summaries) across at least one representative RL setting and one communication setting. This would clarify whether contrastive prompting is essential or simply beneficial.
- **Grasp decomposition clarity:** The grasp experiments rely on a decomposition framework. I would like a clearer breakdown of which parts of the observed gains are attributable to PLLB versus the underlying decomposition and priors. A targeted ablation demonstrating PLLB’s incremental value within this pipeline would strengthen the claim of generality.

---

> ### Author Response · Authors · 2025-12-21
> **Rebuttal**
>
> Thank you for your thoughtful and helpful feedback that has helped us improve the paper!
>
> We address each point under the Requested Changes below:
>
> 1. **Clarifying Pitch**:
>    **We have added a paragraph to Section 3 and a line in the last paragraph of the Introduction** to explicitly clarify that the unifying mechanism behind PLLB is the use of contrastive reward-based rule extraction via an LLM (i.e. positive and negative trajectories to elicit rules), as well as the choice to iteratively condition the policy update on rules. The only variation is in how this conditioning is implemented (Q-regularization vs. prompt conditioning), which is naturally domain specific as it depends on the modality the learning agent operates over.
>
> 2. **Necessity of Language as Bottleneck**:
>    We appreciate the suggestion for isolating language-specific benefits. However, we note a fundamental asymmetry that makes this comparison difficult: non-linguistic bottlenecks (e.g., learned latent codes or contrastive classifiers) would require training on many task variations to learn meaningful abstract representations, whereas the key insight behind PLLB is to leverage the pre-structured abstraction space of LMs to extract useful rules from a single task. This is precisely one of language's key advantages: it provides compositional, generalizable abstractions without task-specific representation learning.
>
>    That said, we believe our existing experiments already provide evidence that the gains are not merely from "any abstraction." The Adversarial ablation shows that corrupted linguistic rules actively hurt performance, demonstrating that rule content matters. Furthermore, our human studies (Sections 6.3, 7.3, 7.4) demonstrate benefits unique to language: interpretability and direct transfer to human collaborators. These properties would be unavailable with latent codes.
>
>    **We have added a discussion of this point to Section 9**, noting that disentangling language-specific benefits from general abstraction would require a substantially different experimental paradigm with several sensitive choices regarding appropriate abstractions.
>
> 3. **The link between PLLB and Rewards**:
>    The concern of PLLB potentially exacerbating reward misspecification is an important point. Precisely studying whether amplification occurs would require environments where we had some perfect reward function to compare with, in which case we would use that instead of the current reward. However, to more fully analyze this point based on our work:
>
>    - **3.1.** Example rules in the Appendix tend to not reference spurious/unrelated attributes, which would be a sign of reward misspecification. For example, our Maze experiment shows that when the reward simply reflects direct task success (solving the maze), PLLB identifies the latent structures (e.g. Red/Blue guidance) without introducing and then amplifying any spurious rules.
>
>    - **3.2.** In SaySelect, PLLB rules actually had a “regularizing” effect versus a potentially misspecified reward, as LLMs had the tendency to output a simple rule of “follow the same action as Agent 1” *even when that wasn’t followed precisely in the prior episodes*. We therefore think it is possible that pre-trained LLMs have a bias *towards* general, simple strategies that may actually reduce, versus enhance, any reward misspecification.
>
>    - **3.3.** While the human subject study in Builder did show some human preference for Baseline rules, we do not view this as a case of reward misspecification as that kind of flexibility was more related to user satisfaction with the overall user study experience (i.e. not having to spend careful time following instructions) rather than anything about the primary task goal of collaborative image reconstruction. However, *we have updated Section 7.3 to clarify this.*
>
>    - **3.4.** One benefit of PLLB is the interpretability of rules. Even if PLLB did amplify reward misspecification, the issue might actually be more detectable and diagnosable, whereas black box policies offer no such transparency.
>
> **We address these points by updating Section 7.3 as well as adding a discussion on reward misspecification in Section 9.**

---

> > ### Author Response · Authors · 2025-12-21
> > **Rebuttal (Cont.)**
> >
> > 4. **Rule Quality and Robustness**:
> > **We have updated Appendix Section B.10 with sensitivity analysis of the SaySelect task**, which we chose as its discrete action space, deterministic dynamics, and multiple optimal solutions provide the clearest signal for detecting sensitivity effects. Concretely, we show that PLLB remains robust to three different kinds of prompt variations, including rephrasing, removing context, and removing format instructions, as well as to different LM temperature parameters when sampling for rule generation.
> >
> >     Regarding the number of contrastive examples, we believe PLLB should use as many contrastive high/low reward examples as possible that fit within the context of the particular LLM. However, we believe it is an interesting direction for future work to design intelligent sampling methods that consider limitations of long-context attention of a given model, and we have updated Section 9 to discuss this.
> >
> >
> > 5. **Contrastive Ablation**:
> > **We have included results for the contrastive ablation for SaySelect (an RL setting) and Birds (a communication setting) in Appendix B.11.** Our results show that contrastive episodes do improve performance of PLLB versus only including high-reward examples, though the latter still does better than our baselines. Furthermore, without example low-reward episodes, the rules are more likely to pick up on spurious signals. For example, with this ablation, image captions in the Birds task were more likely to describe visual features such as a bird with “its mouth wide open” or “looking at the camera,” which are irrelevant to any of our three reward functions. We therefore believe contrasting rules do significantly improve the quality of rules generated by PLLB.
> >
> >
> > 6. **Grasp Decomposition Clarity**:
> > Our Vanilla RL baseline serves as the ablation the reviewer describes and helps isolate PLLB’s contribution by using identical path execution components, solely differing in how the grasp keypoint is selected (RL over image features vs. output from PLLB). **We have updated Sections 8.1 and 8.3 to clarify this.**
> >
> >
> > Thank you for your help in improving our paper!

---

### Author Response · Authors · 2025-12-21
**Overall Rebuttal Response**

Thank you to all reviewers for their helpful feedback! We appreciate that all reviewers found that our findings would be interesting to members of the TMLR community, and that our experiments support the claims made in the submission.

We address each reviewer’s concerns in individual responses. We have completed the following revisions, as requested by reviewers:

- **Clarifying Pitch [h2d6]:** We added a paragraph to Section 3 and a line in the last paragraph of the Introduction to explicitly clarify that the unifying mechanism behind PLLB is the use of contrastive reward-based rule extraction via an LLM, as well as iteratively conditioning an agent’s policy update on rules. The only variation is in how this conditioning is implemented (Q-regularization vs. prompt conditioning), which is naturally domain specific as it depends on the learning agent.

- **Rule Quality and Robustness [h2d6]:** We have added Section B.10 in the Appendix that includes results on sensitivity analysis to prompt variation and sampling temperature for rule generation.

- **Contrastive Ablation [h2d6]:** We have added Section B.11 in the Appendix to show that a non-contrastive ablation that only uses high-reward episodes suffers from lower performance than PLLB, motivating our contrastive approach to rule generation.

- **Updated Limitations and Discussion [all]:** We have updated Section 9 to cover different points raised by reviewers, including investigating the effect of non-linguistic abstractions [h2d6], reward misspecification [h2d6], handling stochastic environments [oXZv], scalability to complex domains [4Yfh], and handling flawed rule generation in safety-critical contexts [4Yfh].

- **Generalization [oXZv]:** We have updated the Introduction, Section 6, and Section 8 to better clarify that our generalization experiments focus on few-shot generalization.

- **Figure 2 [oXZv]:** We have updated Figure 2 to provide more clarity on the details of the diverse set of environments we study.

- **Update Step Clarity [4Yfh]:** We have revised Section 3 to better clarify the update step, and how PLLB can accommodate different implementations of the update based on the underlying learning agent (e.g., RL policy, pure VLM/LLM).

We have also made smaller updates that we describe in the individual responses. We believe our changes address all reviewers’ suggestions, and we appreciate their feedback which helped us improve our paper. We would be happy to answer any further questions.

---

### Decision · Action_Editor_AuDG · 2026-01-29

**Recommendation:** Accept as is

**Audience:**

Yes

**Audience Explanation:**

The reviewers agree that the core idea of using language as a bottleneck to induce abstraction is interesting and worthy of study. They anticipate that the findings will be of interest to many in the TMLR readership.

**Claims And Evidence:**

Yes

**Claims Explanation:**

The reviewers agree that, with the revisions submitted during the review process, the paper's experimental design is sound and its claims are well-supported.